# A cellular and molecular analysis of SoxB-driven neurogenesis in a cnidarian

Eleni Chrysostomou[1†], Hakima Flici[1‡], Sebastian G Gornik[1§], Miguel Salinas-Saavedra[1#], James M Gahan[1], Emma T McMahon[1], Kerry Thompson[2], Shirley Hanley[3], Michelle Kilcoyne[4], Christine E Schnitzler[5,6], Paul Gonzalez[7], Andreas D Baxevanis[7], Uri Frank[1]*

[1]Centre for Chromosome Biology, School of Natural Sciences, National University of Ireland Galway, Galway, Ireland; [2]Centre for Microscopy and Imaging, Discipline of Anatomy, National University of Ireland, Galway, Galway, Ireland; [3]National Centre for Biomedical Engineering Science, National University of Ireland, Galway, Galway, Ireland; [4]Carbohydrate Signalling Group, Microbiology, School of Natural Sciences, National University of Ireland Galway, Galway, Ireland; [5]Whitney Laboratory for Marine Bioscience, University of Florida, St. Augustine, Florida, United States; [6]Department of Biology, University of Florida, Gainesville, Florida, United States; [7]Computational and Statistical Genomics Branch, Division of Intramural Research, National Human Genome Research Institute, National Institutes of Health, Bethesda, Maryland, United States

*For correspondence:
uri.frank@nuigalway.ie

Present address: [†]Univ Paris Est Creteil, INSERM, IMRB, Creteil, France; [‡]Institut de Génétique et de Biologie Moléculaire et Cellulaire (IGBMC), Strasbourg, France; [§]Centre for Organismal Studies (COS), Heidelberg University, Heidelberg, Germany; [#]Sars Centre for Marine Molecular Biology, University of Bergen, Bergen, Norway

Competing interest: The authors declare that no competing interests exist.

**Abstract** Neurogenesis is the generation of neurons from stem cells, a process that is regulated by SoxB transcription factors (TFs) in many animals. Although the roles of these TFs are well understood in bilaterians, how their neural function evolved is unclear. Here, we use *Hydractinia symbiolongicarpus*, a member of the early-branching phylum Cnidaria, to provide insight into this question. Using a combination of mRNA in situ hybridization, transgenesis, gene knockdown, transcriptomics, and in vivo imaging, we provide a comprehensive molecular and cellular analysis of neurogenesis during embryogenesis, homeostasis, and regeneration in this animal. We show that SoxB genes act sequentially at least in some cases. Stem cells expressing *Piwi1* and *Soxb1*, which have broad developmental potential, become neural progenitors that express *Soxb2* before differentiating into mature neural cells. Knockdown of SoxB genes resulted in complex defects in embryonic neurogenesis. *Hydractinia* neural cells differentiate while migrating from the aboral to the oral end of the animal, but it is unclear whether migration per se or exposure to different microenvironments is the main driver of their fate determination. Our data constitute a rich resource for studies aiming at addressing this question, which is at the heart of understanding the origin and development of animal nervous systems.

## Editor's evaluation

This paper shows that SoxB genes act sequentially in neural stem cells before differentiating into mature neural cells and that loss of SoxB genes causes defects in embryonic neurogenesis. The manuscript provides molecular insights into neurogenesis during cnidarian embryogenesis, homeostasis, and regeneration.

## Introduction

Neurogenesis – the generation of neuronal cells – is thought to be a partially conserved process in metazoans (*Bosch et al., 2017*; *Gahan et al., 2017*; *Kelava et al., 2015*; *Rentzsch et al., 2017*; *Tournière et al., 2020*; *Watanabe et al., 2014*) but the early evolutionary history of this process

remains unclear. SoxB transcription factors (TFs) are central players in neurogenesis across a broad range of animal taxa including vertebrates, insects, nematodes, planarians, and cnidarians (*Alqadah et al., 2015*; *Meulemans and Bronner-Fraser, 2007*; *Richards and Rentzsch, 2015*; *Ross et al., 2018*) but how and when these genes became co-opted into the process of neuron formation is unknown. Vertebrate SoxB TFs are commonly subdivided into the SoxB1 group (Sox1, Sox2, and Sox3) that maintain stemness in neural stem cells, while SoxB2 group members (Sox14 and Sox21) are involved in neural differentiation (*Bylund et al., 2003*; *Holmberg et al., 2008*). Hence, vertebrate SoxB genes act sequentially in neurogenesis, but it is unclear whether this feature is ancestral or derived in metazoan animals. Furthermore, SOX orthology has been notoriously hard to interpret and, therefore, a similar SoxB subdivision has not been observed in invertebrates (*Flici et al., 2017*; *Jager et al., 2011*; *Richards and Rentzsch, 2015*; *Schnitzler et al., 2014*). Finally, expression and functional data on SOX genes in the context of neurogenesis are still rare in non-bilaterians, making an evolutionary inference difficult to establish.

Cnidarians are the sister group to bilaterians (*Kayal et al., 2018*; *Zapata et al., 2015*) and, as such, they represent an interesting outgroup to study the evolution of neurogenesis. As in vertebrates, multiple SoxB genes are expressed in the developing nervous systems of representatives of the two main cnidarian clades, Anthozoa (corals and sea anemones) (*Magie et al., 2005*; *Richards and Rentzsch, 2014*; *Richards and Rentzsch, 2015*) and Medusozoa (hydroids and jellies) (*Flici et al., 2017*; *Jager et al., 2006*). While we know that some cnidarian SoxB genes are required for neurogenesis, their function is not well understood. *Hydractinia symbiolongicarpus* is a colonial hydrozoan cnidarian (*Figure 1—figure supplement 1*; *Frank et al., 2020*) and, like other hydrozoans, this animal possesses a population of stem cells, known as i-cells. These i-cells self-renew and contribute to somatic lineages, including neurons, and to germ cells throughout life (*DuBuc et al., 2020*; *Gahan et al., 2016*). The colonial nature of the animal enable i-cells to migrate throughout the colony and populate individual polyps. Hence, i-cells constitute a single population for the entire colony. i-cells first appear in embryonic endoderm (*Gahan et al., 2016*) but migrate to the outer tissue layer of the body, the epidermis, during metamorphosis (*Figure 1—figure supplement 2*), remaining there during adult life. In the mature animal, undifferentiated i-cells are found in interstitial spaces of the epidermis, primarily of the lower body column. They differentiate while migrating orally (*Bradshaw et al., 2015*), giving rise to various cell types including the two main neuronal lineages: neurons and nematocytes – the latter being specialized stinging cells that are used by cnidarians to capture prey or for defence from predators. The *Hydractinia* nervous system has a typical cnidarian nerve-net structure with a higher concentration of neurons at its oral end (*Figure 1—figure supplement 3* and *Figure 1—figure supplement 4*). The *Hydractinia* genome encodes three SoxB-like genes (*Soxb1*, *Soxb2*, and *Soxb3*) whose orthologous relation to specific bilaterian homologs cannot be resolved due to low sequence conservation. Hence, their names do not implicate them as being part of any bilaterian SoxB subgroup (*Flici et al., 2017*). A previous study has shown that *Hydractinia Soxb2* is expressed in proliferative, putative neural progenitors, and *Soxb3* is expressed in neural cells, identified by their distinctive morphology. Both genes are essential for adult neurogenesis and nematogenesis (*Flici and Frank, 2018*; *Flici et al., 2017*). However, the functions of *Soxb1*, *Soxb2*, and *Soxb3* in embryonic neurogenesis remain unknown.

Here, we have studied the function of all three *Hydractinia* SoxB genes in embryogenesis. To observe the dynamic expression of these genes in adults, we have generated *Soxb1* and *Soxb2* transgenic reporter animals and tracked individual cells during differentiation using in vivo time lapse imaging. We also used fluorescence-activated cell sorting (FACS) and droplet microfluidics approaches (InDrop-seq) as a first step towards generating cell type-specific and single-cell transcriptomes, then analysed the cell cycle characteristics of *Hydractinia* cell fractions. Our data are consistent with SoxB genes acting sequentially in adults as i-cells differentiate into neurons and nematocytes. Knockdown experiments indicate a complex role for SoxB genes in the differentiation of embryonic neuronal cells. Under *Soxb2* knockdown conditions, we have documented biased differentiation to specific neural cell types. Our study provides insight into the evolution of SoxB genes and their function in neurogenesis across Metazoa.

## Results

### Expression pattern of SoxB genes in *Hydractinia*

We studied the expression pattern of all three SoxB genes in *Hydractinia* using both traditional, tyramide amplification-based mRNA fluorescence in situ hybridization (FISH) and signal amplification by exchange reaction (SABER) single-molecule FISH (*Kishi et al., 2019*). In sexual polyps, *Soxb1* was expressed in cells morphologically resembling i-cells and germ cells in males and females (*Figure 1A–D*). In feeding polyps, *Soxb1* was expressed in cells that resembled i-cells both with respect to their morphology and their anatomical location, being primarily distributed in the lower body column of the polyp and largely absent from the oral end of the animal (*Bradshaw et al., 2015*; *Figure 1E–G*). To confirm the identity of *Soxb1*-expressing cells, we used double FISH with *Piwi1*, a known marker for *Hydractinia* i-cells and germ cells (*DuBuc et al., 2020*; *Gahan et al., 2016*). We found that all *Soxb1*⁺ cells also expressed *Piwi1* and vice versa (*Figure 1*). Therefore, we conclude that *Soxb1* is expressed in i-cells, that is, in stem cells that give rise to somatic cells (including neurons and nematocytes) and to germ cells. *Soxb2* expression partially overlapped with *Soxb1* in the mid-region of the polyp body column, being largely absent from the lower (aboral) area of the polyp that primarily included only *Piwi1*⁺/*Soxb1*⁺ cells (*Figure 2A–C*). *Soxb3* was expressed in the middle and upper (i.e., oral) parts of the polyp and partly overlapped with *Soxb2* in the mid-region (*Figure 2D–F*). These findings are consistent with a previous study that investigated *Soxb2/Soxb3* expression (*Flici et al., 2017*). It is also in line with the known pattern of distribution of *Hydractinia* cells, with i-cells being restricted to the lower body column, and neural cells mainly concentrated in the oral pole.

While mRNA FISH experiments, described above, informed us about cells that expressed SoxB genes in space and time at cellular resolution, they did not provide direct information on the fate of these cells. To obtain these data, we generated transgenic reporter animals that expressed tdTomato or GFP, driven by the endogenous *Soxb1* or *Soxb2* genomic control elements, respectively (*Figure 3A–I*). Attempts to generate a *Soxb3* reporter were unsuccessful. Instead, we used an *Rfamide* reporter animal that expresses GFP in RFamide⁺ neurons (*Figure 3J–L*). The rationale of this approach was that the long half-life of fluorescent proteins would enable tracking cells even after their fate is changed and the reporter gene shut down. Fluorescence of the *Soxb1::tdTomato* reporter was dim in vivo but the signal was enhanced in fixed animals through the use of anti-RFP antibodies (*Figure 3A–E*). In contrast, the signal produced by the *Soxb2* and *Rfamide* reporters, both expressing GFP, was sufficiently bright to be viewed in vivo using fluorescence stereomicroscopy (*Figure 3F–L*).

Contrary to the mRNA FISH data (*Figure 1*), the *Soxb1* promoter-driven tdTomato was not only observed in stem/progenitor cells but was present in neural cells as well, based on the distinct morphology of these cells (*Figure 3A–C*). In sexual polyps, tdTomato was also observed in maturing germ cells (*Figure 3D and E*). This shows that *Soxb1*⁺ cells differentiated into neural cells and gametes as expected. The *Soxb2* promoter-driven GFP was visible in neurons and nematocytes but not in i-cells (*Figures 2 and 3F–I*), suggesting that *Soxb2*⁺ cells were more committed than i-cells (i.e., *Soxb1*⁺ cells). The spatial distribution of *Soxb1*⁺ and *Soxb2*⁺ cells was distinct from that of their respective progeny, consistent with migrating neural progenitors that differentiate during and following their migration away from the lower body column area where i-cells and their early progeny reside. The *Rfamide* reporter was only visible in sensory and ganglionic neurons (*Figure 3J–L*), consistent with the expression pattern of this gene in neurons but not in neural progenitors (*Gahan et al., 2017*; *Kanska and Frank, 2013*).

FISH expression patterns (*Figure 2*) showed partial overlap between *Soxb1* and *Soxb2*, as well as between *Soxb2* and *Soxb3*, the latter confirming previous reported observations (*Flici et al., 2017*). Furthermore, analysis of transgenic reporter animals indicated that *Soxb1*⁺ and *Soxb2*⁺ cells give rise to neural cells (*Figures 2 and 3*). To confirm within-lineage transitions between *Soxb1* and *Soxb2* expression, we generated double *Soxb1::tdTomato/Soxb2::GFP* reporter animals by crossing a *Soxb1::tdTomato* female with a *Soxb2::GFP* male. The *Soxb1*-driven tdTomato fluorescence that was too dim to be viewed live by stereomicroscopy was readily visible in vivo using a more sensitive spinning disk confocal microscope. Double transgenic animals were mounted in low-melt agarose and subjected to prolonged in vivo spinning disk confocal imaging. We observed tdTomato-positive (i.e., *Soxb1*⁺), GFP-low or negative (i.e., *Soxb2*⁻) cells transforming to double positive cells (*Figure 4*; *Figure 4—figure supplement 1*). Collectively, our mRNA expression data and the ones obtained previously (*Flici et al.,*

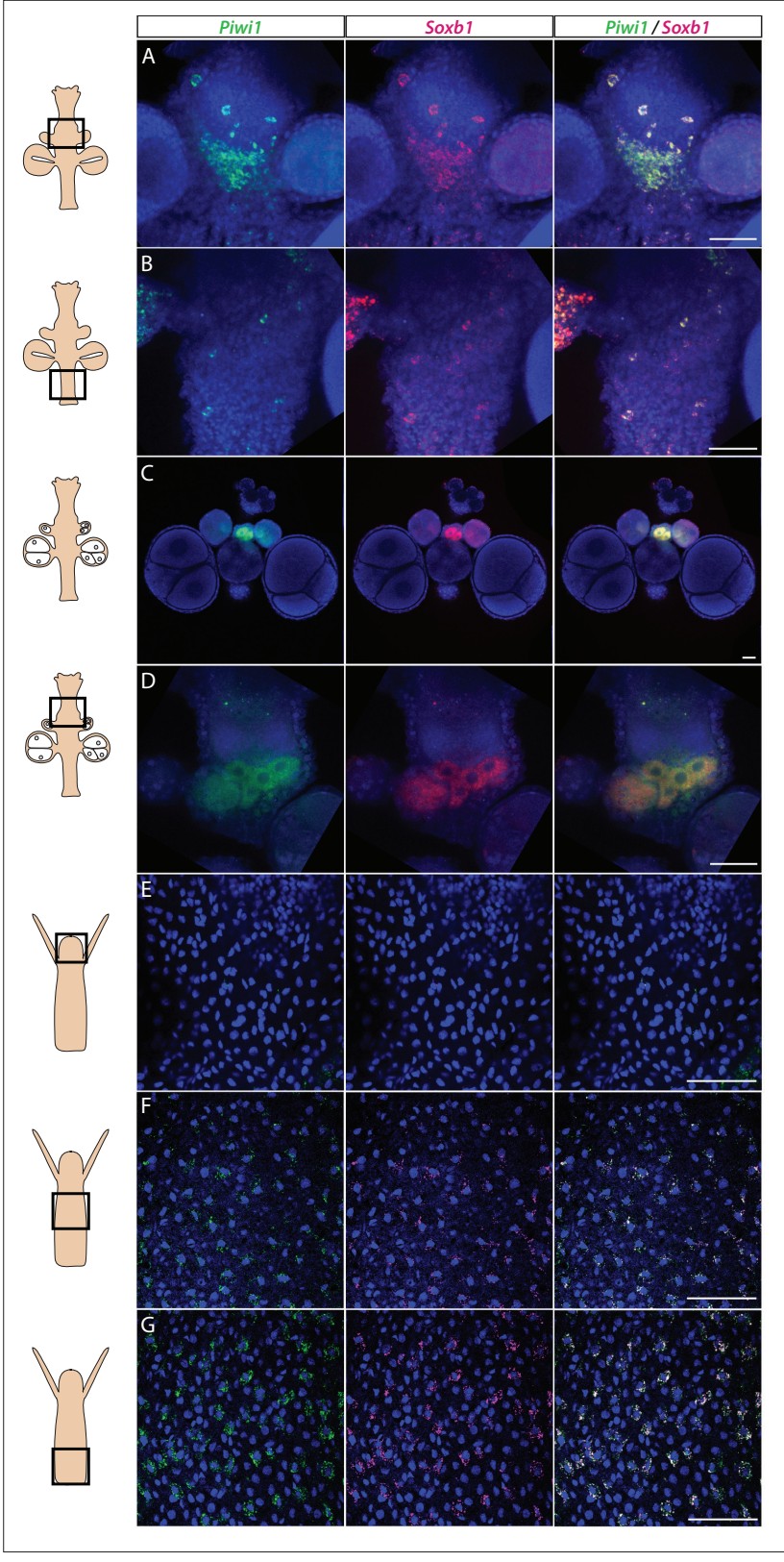

**Figure 1.** *Piwi1* and *Soxb1* are co-expressed in i-cells and germ cells. (**A–D**) Analysis of sexual polyps' mid (**A**) and lower (**B**) body column, and early (**C**) and late (**D**) oocytes. (**E–G**) Analysis of feeding polyps' head (**E**), mid-(**F**), and lower (**G**) body column. (**A–D**) were tyramide-based fluorescence in situ hybridization (FISH); (**E–G**) were single-molecule FISH. Scale bars = 40 µm.

*Figure 1 continued on next page*

*Figure 1 continued*

The online version of this article includes the following figure supplement(s) for figure 1:

**Figure supplement 1.** The animal model *Hydractinia.*

**Figure supplement 2.** i-Cell location during different stages in ontogeny, marked by Piwi1.

**Figure supplement 3.** The structure of the nervous system in embryogenesis and larvae.

**Figure supplement 4.** The structure of the nervous system in adult feeding and sexual polyps.

*2017*) together with the results of our live imaging of transgenic reporter animals are consistent with sequential expression of SoxB genes in the neural lineage.

## Cellular and molecular characterization of *Hydractinia* neural cells

We used a recently developed flow cytometry/FACS protocol (*DuBuc et al., 2020*) to analyse and physically sort cells from dissociated transgenic reporter animals. The long half-life of the fluorescent reporter proteins resulted in the isolation of not only cells that expressed the reporter genes but also their progeny. To minimize the contribution of progeny to the sorted cellular fraction, we used a FACS gating strategy to select live, nucleated single cells and sorted populations that had low side scatter (SSC) characteristics and high levels of GFP (*Figure 5A–C* and *Figure 5—figure supplement 1*).

A previously established *Piwi1::GFP* reporter animal (*Bradshaw et al., 2015*) was used to selectively isolate i-cells by FACS rather than the *Soxb1::tdTomato* reporter animal. tdTomato fluorescent protein is optimally excited by a 561 nm yellow/green laser; the configuration of the BD FACS Aria cell sorter in the NUI Galway flow cytometry core facility does not include a 561 nm laser. The usage of *Piwi1::GFP* to isolate i-cells was possible given that the two genes have been shown to be co-expressed in i-cells (*Figure 1*). $Piwi1^+$ (i.e., $Soxb1^+$ i-cells) are known to represent a rare population, representing approximately 1.4% of the live single-cell population of the *Soxb2::GFP* transgenic animal (*DuBuc et al., 2020*). We found that $Soxb2^+$ cells (i.e., putative neural progenitors) were an equally rare population (*Figure 5A–B*). $Rfamide^+$ neurons were distributed among two distinct populations that differed in levels of GFP expression, accounting for 1.6% and 1.8% of the live single cells, respectively (*Figure 5C*). Imaging flow cytometry of the *Piwi1*::GFP reporter animal (i.e., cells that also express *Soxb1*) revealed that the brightest cells were small with a high nuclear to cytoplasmic ratio, consistent with stem cell morphology (*Figure 5D*). The two $Rfamide^+$ cell populations did not differ morphologically based on imaging flow cytometry except in their GFP expression (*Figure 5E and F*); their exact nature is yet unclear. Flow cytometric cell cycle analysis (*Figure 5G–L*) revealed that the $Piwi1^+$ cells were almost exclusively in S/G2/M phase (*Figure 5G*), similar to what has been observed in i-cells from the freshwater polyp *Hydra* (*Buzgariu et al., 2014*). However, i-cell progeny, defined by low GFP fluorescence and constituting a highly heterogeneous population, were found in all stages of the cell cycle (*Figure 5H*). $Soxb2^+$ cells that include putative neural progenitors (*Flici et al., 2017*) had a cell cycle profile similar to i-cells, mostly found during the S/G2/M phases of the cell cycle (*Figure 5J*). $Rfamide^{high}$ and $Rfamide^{low}$ neurons that are thought to be terminally differentiated were mostly in G0/G1, similar to what was observed in nematocytes (*Figure 5I and K–L*). The robustness of the flow cytometric analysis was demonstrated by studying other cellular fractions including epithelial cells from an epithelial reporter (*Künzel et al., 2010*), male germ cells (from the *Piwi1* reporter), and cells from an *Ef1a* knock-in animal where GFP is expressed ubiquitously (*Sanders et al., 2018*; *Figure 5—figure supplement 1* and *Figure 5—figure supplement 2*).

We used FACS to physically isolate the following cell fractions: $Soxb2^{high}$, $Rfamide^{high}$, and $Rfamide^{low}$ neurons. We generated transcriptomes for each of these cell fractions and used a previously published i-cell transcriptome (*DuBuc et al., 2020*). We then compared gene expression between the combinations shown in *Supplementary file 1*. Given the long half-life of fluorescent proteins, the $Soxb2^+$ cells' transcriptome we present (*Supplementary file 2*) probably includes also early stages of their differentiation. The two $RFamide^+$ neuronal populations are probably different, as the RFamide neuropeptide is considered a marker for differentiated neurons. Indeed, typical neuronal genes were expressed by these cells (*Figure 5M*; *Supplementary file 2*). The $RFamide^+$ neurons fraction also expressed minicollagen genes that are markers of nematogenesis. Co-expression of minicollagens and precursors of neuropeptides has been reported previously (*Sunagar et al., 2018*; *Wolenski et al., 2013*) so this observation in $RFamide^+$ cells is not surprising given that nematocytes are neural cells.

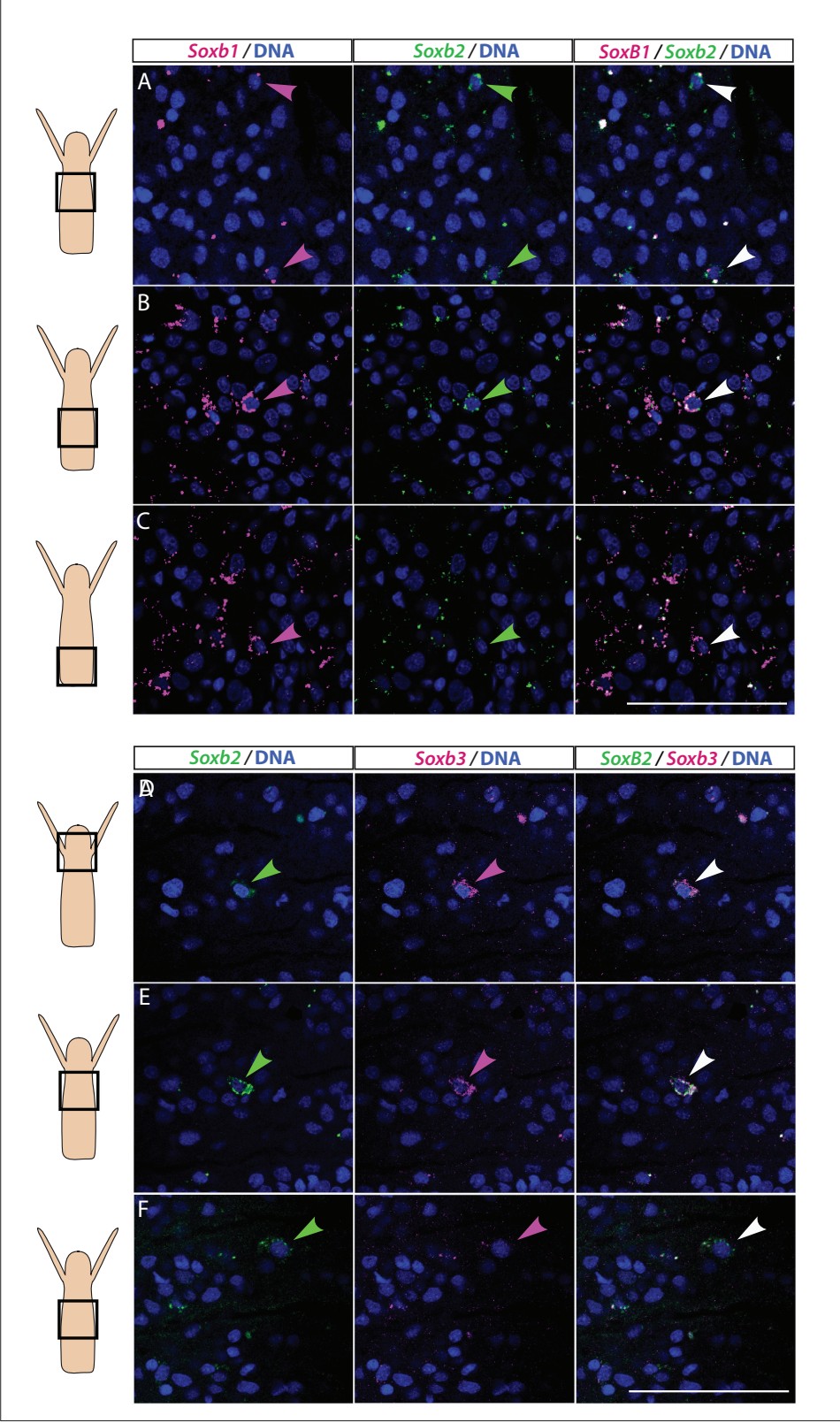

**Figure 2.** Partial overlap in the expression of *Soxb1* and *Soxb2*, and *Soxb2* and *Soxb3* in feeding polyps. (**A–C**) Single-molecule fluorescence in situ hybridization (FISH) with probes against *Soxb1* and *Soxb2 showing* upper-mid (**A**), lower-mid (**B**), and lower (**C**) body column. (**D–F**) Single-molecule FISH with probes against *Soxb2* and *Soxb3* showing the head region (**D**), upper-mid (**E**), and lower-mid (**F**) body column. Arrowheads point to double-positive cells. Scale bars = 40 μm.

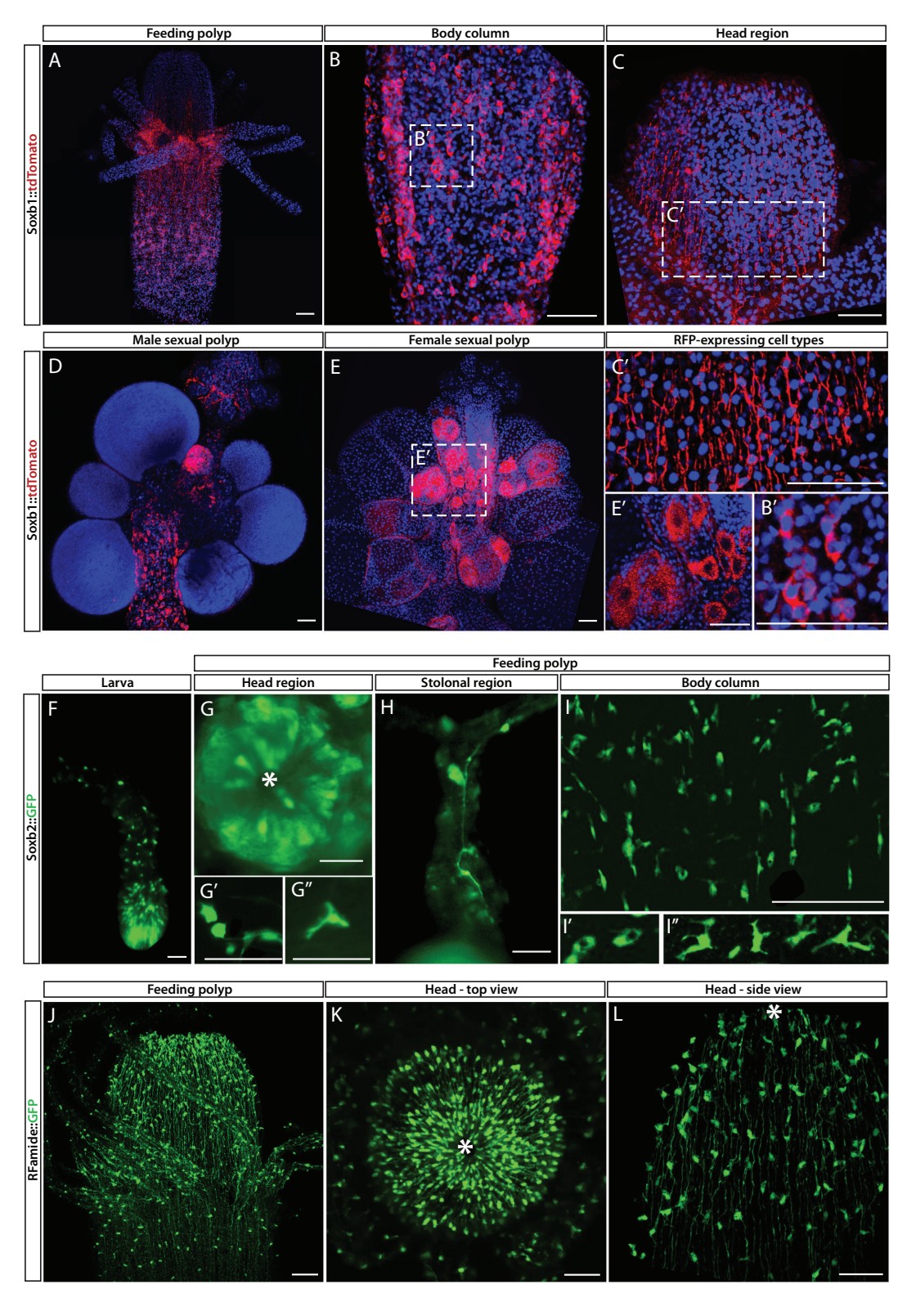

**Figure 3.** *Soxb1, Soxb2,* and *Rfamide* transgenic reporter animals. (**A–E**) *Soxb1::tdTomato* reporter animal. Animals were fixed and stained with an anti-dsRed antibody. (**A**) Whole feeding polyp. (**B**) Lower polyp body column. (**B'**) Higher magnification of tdTomato⁺ i-cells. (**C**) Head region including base of tentacles. (**C'**) Close-up showing tdTomato⁺ neurites. (**D**) Male sexual polyp. (**E**) Female sexual polyp. (**E'**) Higher magnification of tdTomato⁺ oocytes. (**F–I**) Live imaging of a *Soxb2::GFP* reporter animal. (**F**) planula larva. (**G**) Oral pole of a polyp viewed from above. (**G', G''**) Higher magnification

*Figure 3 continued on next page*

*Figure 3 continued*

of GFP⁺ cells with neural morphology. (**H**) GFP⁺ interconnected neurons in stolon. (**I**) Mid-body column region of polyp showing GFP⁺ cells. (**I′**) GFP⁺ nematoblasts (based on the presence of a nematocyst capsule). (**I″**) GFP⁺ neurons. (**J–L**) Live imaging of *Rfamide::GFP*⁺ reporter feeding polyps. (**J**) Upper polyp region including head and tentacles. (**K**) Oral pole of feeding polyp viewed from above. (**L**) Higher magnification of the head region, showing GFP⁺ neural network. Asterisks point to oral ends. Scale bars = 40 µm.

To provide further insight into the composition of the neural cells population, we used a droplet microfluidic single-cell handling system (InDrop) (*Klein et al., 2015*) to generate a single-cell transcriptome dataset based on 7071 cells from *Hydractinia* wild type feeding polyps. This allowed us to identify 22 cell clusters (*Figure 5—figure supplement 3A*) and their associated upregulated marker genes (*Figure 5—figure supplement 3B*); this may be an underestimation of the cellular complexity of *Hydractinia* as compared to that of other cnidarians (*Sebé-Pedrós et al., 2018*; *Siebert et al., 2019*). However, we were able to identify putative i-cells, nematoblasts, gland cells, neurons, and a group of mixed progenitors. Upregulated marker gene expressions generally recapitulated the expression profiles of the physically sorted cells from the transgenic reporter animals, with up to 15% of markers corresponding to DEG transcript of the sorted cell types (*Figure 5—figure supplement 3C*; *Supplementary file 3*). Specifically analyzing the expression of *Soxb1-3* (*Figure 5—figure supplement 3D*), we find that both *Soxb1* and *Soxb2* are expressed in a subset of i-cells; these cells are presumably in

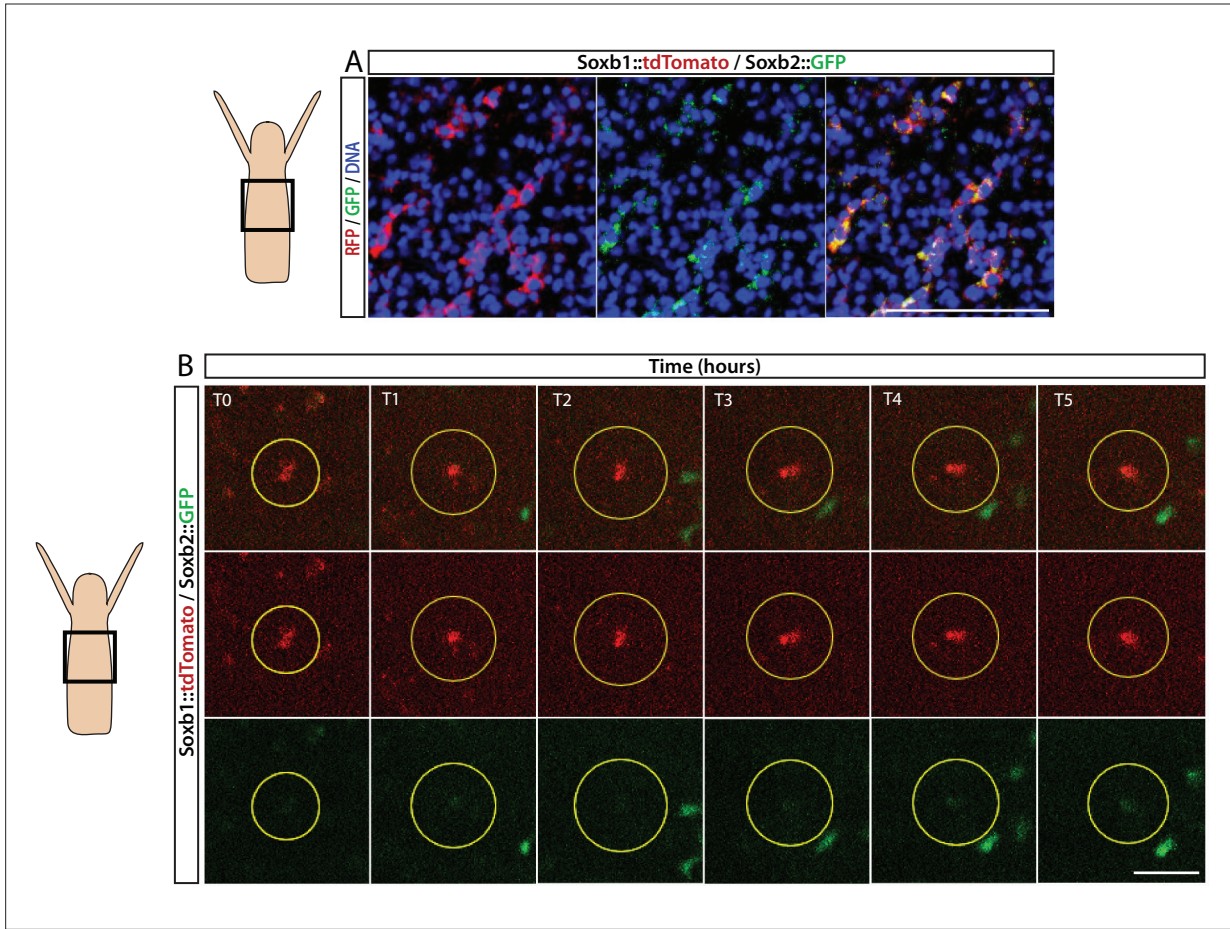

**Figure 4.** Sequential expression of *Soxb1* and *Soxb2*. (**A**) Immunofluorescence of *Soxb1::tdTomato* and *Soxb2::GFP* double positive cells that resemble differentiating neurons by morphology. (**B**) In vivo time lapse imaging of a *Soxb1::tdTomato*⁺ cells (shown in red) with increasing levels of *Soxb2::GFP* (shown in green) over a time frame of 8 hr (T0=3 hr post decapitation (hpd); T1=4 hpd;...; T5=8 hpd). Green *Soxb2::GFP* cells that appear in the image probably migrated into the confocal plane during imaging. Scale bars = 40 µm.

The online version of this article includes the following figure supplement(s) for figure 4:

**Figure supplement 1.** In vivo time lapse imaging of a *Soxb1::tdTomato/Soxb2::GFP* double transgenic reporter animal.

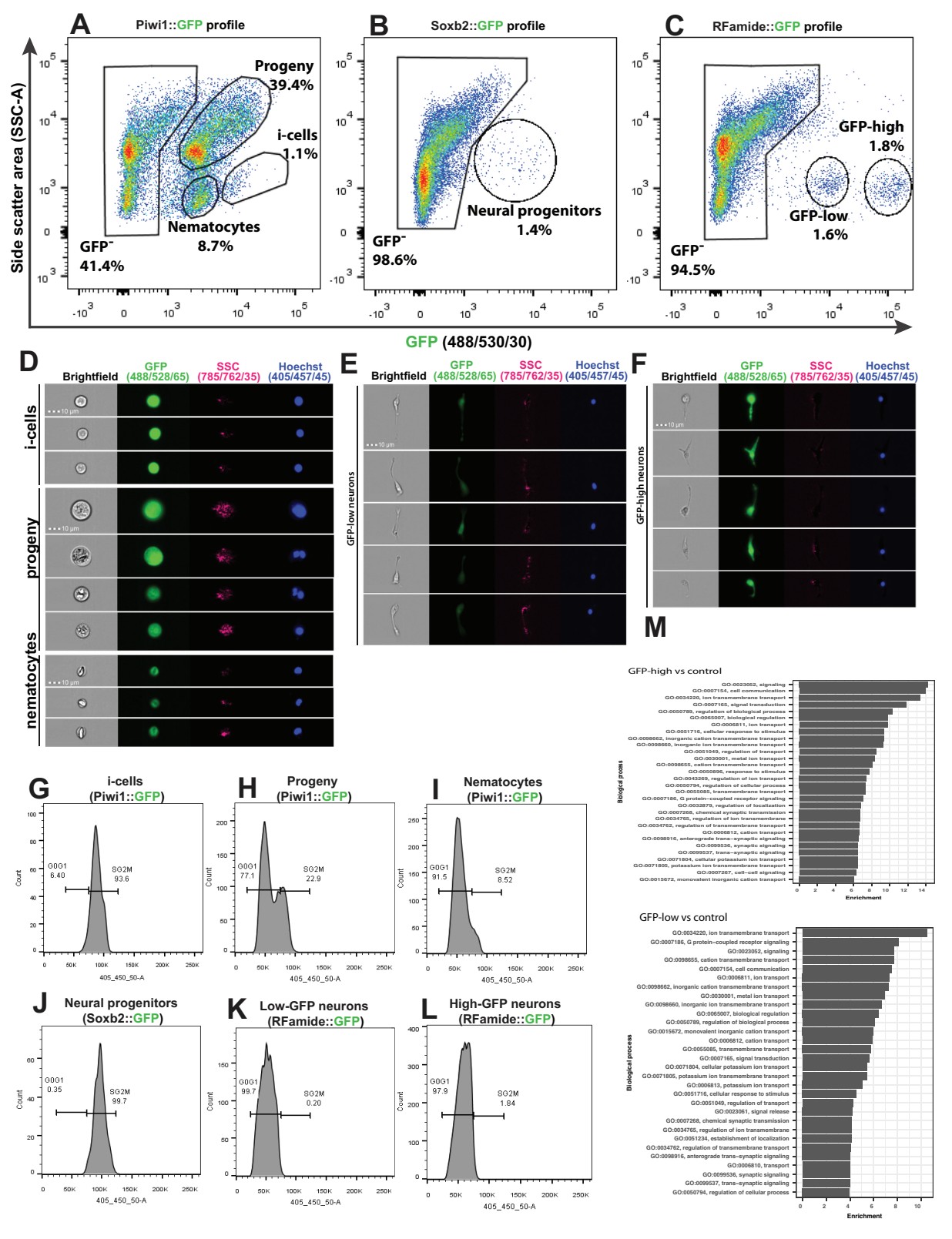

**Figure 5.** Analysis of dissociated *Hydractinia* cells by conventional and imaging flow cytometry. (**A**) Gating strategy of i-cells, nematocytes, mixed progenitors, and GFP⁻ cells from a *Piwi1* reporter animal. (**B**) Gating of putative neural progenitors from a *Soxb2* reporter animal. (**C**) Gating two distinct populations of *Rfamide*::GFP⁺ neurons. (**D–F**) Imaging flow cytometric analysis of cells from the transgenic reporter animals. (**G–L**) Cell cycle analysis of distinct, sorted cell populations. (**G**) i-Cells are mostly in S/G2/M. (**H**) i-Cell progeny are distributed along the cell cycle. (**I**) Nematocytes are mostly in G1.

*Figure 5 continued on next page*

*Figure 5 continued*

(**J**) *SoxB*+ (putative neural progenitors) are mostly in S/G2/M. (**K**) Low and (**L**) high *Rfamide*+ neurons. (**M**) Differential gene expression analysis of sorted cell populations. Genes expressed by *Rfamide*-high and *Rfamide*-low neurons are listed.

The online version of this article includes the following figure supplement(s) for figure 5:

**Figure supplement 1.** Representative flow cytometry density plots identifying subpopulations as defined by internal complexity (side scatter [SSC]) and level of GFP expression of *Hydractinia* transgenic reporter animals and matched wild type animals (**A–I**) and imaging flow cytometry (**J**).

**Figure supplement 2.** Typical flow cytometry gating strategy used for cell cycle analysis of transgenic animals and a wild type animal.

**Figure supplement 3.** Transcriptomic analysis of Hydractinia single cells.

transition to becoming *Soxb2*-only positive progenitor cells (clusters 13 and 14 in *Figure 5—figure supplement 3A*). Finally, *Soxb3* is only expressed in few cells in cluster 18 (*Figure 5—figure supplement 3A, D*). It also appears that subpopulations of neural cells transition from a *Soxb2* RFamide low state (cluster 14) that express a G-protein coupled receptor for glutamate (*Figure 5—figure supplement 3E*) to a *Soxb2* RFamide high state (cluster 13), of which some eventually become *Soxb2*-negative while still expressing high levels of RFamide (cluster 11). These are presumably terminally differentiated neurons as determined by the marker genes Kaptin and Contactin2 (*Figure 5—figure supplement 3E*). This is consistent with the downregulation of *Soxb2* resulting in near absence of RFamide+ neurons and bolsters the role of *Soxb2* as a marker for neural progenitor cells. Combined, the bulk- and single-cell analyses provide a valuable resource for further analyses of cnidarian cells, allowing for the extended analysis of nematoblasts as a way to characterize both known and novel marker genes (*Figure 5—figure supplement 3F*).

### *Hydractinia* head regeneration involves de novo neurogenesis

*Hydractinia* can regenerate a decapitated head that includes a fully functional nervous system within 2–3 days (*Bradshaw et al., 2015*). This predictable generation of many new neurons provides an opportunity to study neurogenesis in regeneration. To identify the cellular source of regenerative neurogenesis, we first characterized the dynamics of nervous system regeneration over 48 hr using *Rfamide::GFP* reporter transgenic animals (*Figure 6*). Animals were decapitated and allowed to regenerate their heads. Subsets of these animals were then fixed at 12, 24, 36, and 48 hr post decapitation (hpd). Prior to fixation, they were incubated for 30 min in EdU. Fixed samples were stained for EdU, GFP (reporting *Rfamide*), and Piwi2 (an i-cell marker; for anti-Piwi2 antibody validation; see *Figure 6—figure supplement 1*). We also analysed intact animals that showed a well-developed head nervous system but had neither S-phase cells nor i-cells in their heads (*Figure 6A*), as expected (*Bradshaw et al., 2015*). At 12 hpd, there were many cycling cells in the stump area, few neurons (likely remnants of the amputated tissue), and few migrating Piwi2low cells that are probably i-cell progeny (*Figure 6B*). At 24 hpd, a clear proliferative blastema had been established at the oral most pole, with i-cells (Piwi2high) and their progeny being numerous in the area. However, only a few RFamide+ neurons were visible (*Figure 6C*). At 36 hpd, new neurons were visible in the regenerating head and proliferative i-cells and progeny were still present (*Figure 6D*). Finally, at 48 hpd, a nearly complete head nervous system had been established and the regenerating head contained lower numbers of i-cells and progeny (*Figure 6E*). Hence, Piwi2low cells, which are presumably i-cell progeny, were the first to migrate to the injury area, followed later on by Piwi2high cells, which are probably stem cells. This is similar to what has been observed in planarian head regeneration, where neoblast progeny arrive in the injury area before neoblasts (*Abnave et al., 2017*). Of note, none of the existing neurons incorporated EdU during regeneration, consistent with neurons being terminally differentiated. To study nervous system regeneration in vivo, we used the *Rfamide*::GFP transgenic reporter animals to track individual neurons over the course of regeneration. The animals were decapitated, mounted in low-melt agarose, and imaged every hour for either the first 24 hr, or from 24 to 72 hpd. In all cases, the neurons remained stationary and did not proliferate during the observation period while a new head nervous system regenerated (*Figure 6—figure supplement 2*). Hence, similar to *Hydra* (*Miljkovic-Licina et al., 2007*), head nervous system regeneration in *Hydractinia* involves de novo neurogenesis rather than proliferation or migration of existing neurons.

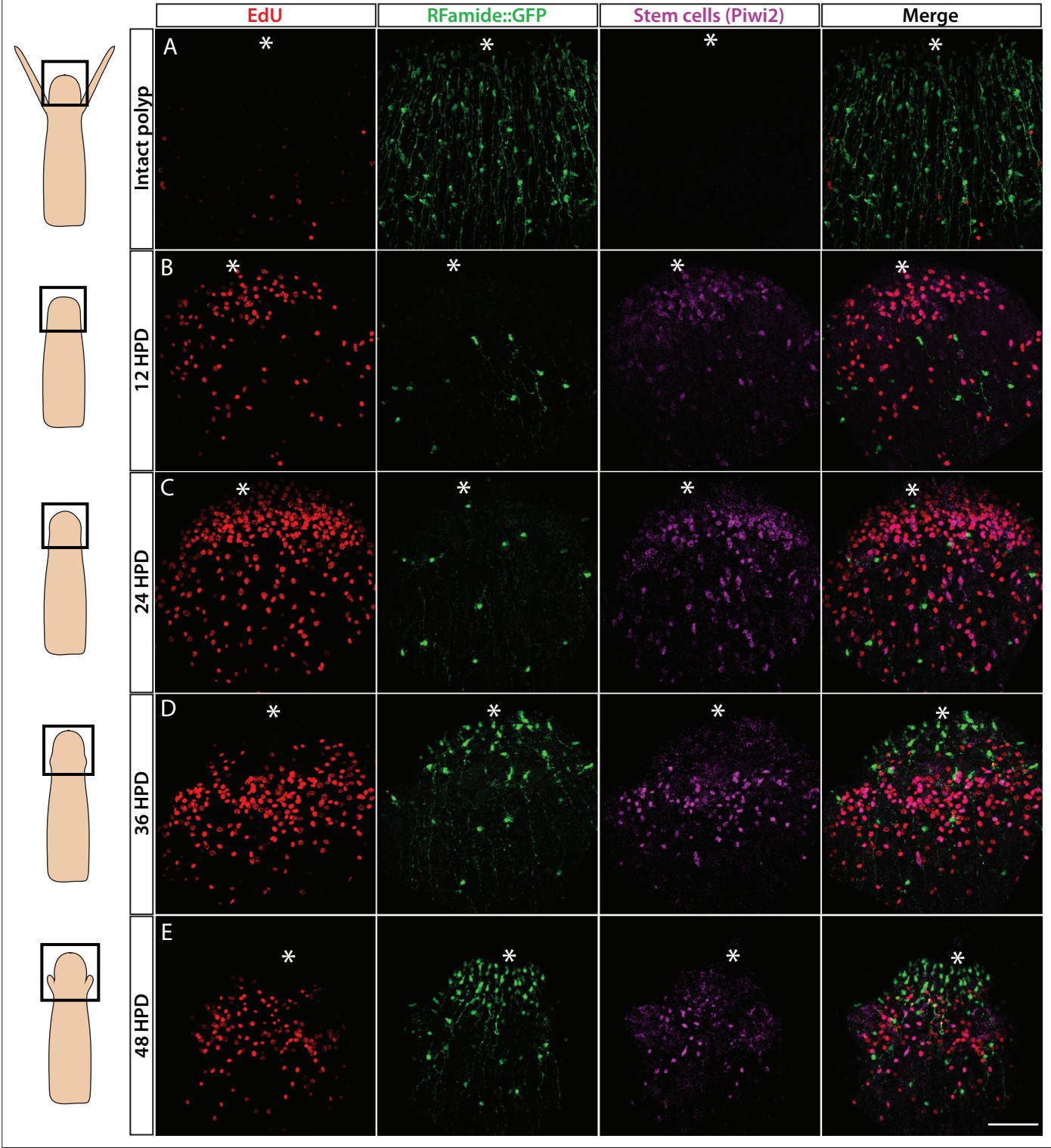

**Figure 6.** Cellular events during *Hydractinia* head regeneration in an Rfamide::GFP transgenic reporter animal. (**A**) Intact head, characterized by few S-phase cells, extensive RFamide+ neuronal network, and no Piwi1high i-cells or their putative Piwi1low progeny. (**B**) 12 hr post decapitation (hpd) showing many proliferative cells, only few remnant RFamide + neurons, and few immigrating i-cells and progeny. (**C**) 24 hpd. High numbers of EdU incorporating cells, few RFamide+ neurons, and increasing i-cells and putative progeny are seen at the oral regenerating end. (**D**) 36 hpd. High number of S-phase cells, increased number of RFamide+ neurons, and further increase in i-cell numbers are seen. (**E**) 48 hpd. Decreasing numbers of S-phase cells, increasing numbers of RFamide+ neurons, and decreasing numbers of i-cells are observed. Scale bars = 40μm.

*Figure 6 continued on next page*

*Figure 6 continued*

The online version of this article includes the following figure supplement(s) for figure 6:

**Figure supplement 1.** Piwi2 antibody validation.

**Figure supplement 2.** In vivo tracing of RFamide-GFP⁺ neurons during regeneration located in the (**A**) lower part and (**B**) upper part of the body column during the first 20 hr post decapitation (hpd), and (**C**) from 24 hpd until 72 hpd.

## SoxB genes are required for embryonic neurogenesis

It has been shown previously that *Soxb2* and *Soxb3* are essential for neurogenesis in adult *Hydractinia* (***Flici et al., 2017***). However, the role of *Soxb1* has not been studied, nor has the role of these TFs in other life stages. Hence, we set out to study the role of all three SoxB genes in embryogenesis. For this, we used short hairpin RNA (shRNA)-mediated knockdown by microinjection into zygotes as previously described (***DuBuc et al., 2020***). The specificity of the treatment was assessed qualitatively by FISH in at least three replicates per shRNA (***Figure 7—figure supplement 1A***). The animals were fixed at 72 hpf; at this stage of development, *Hydractinia* normally reaches the metamorphosis-competent planula larva stage.

*Soxb1* knockdown affected many aspects of larval development. First, the animals were on average smaller and underdeveloped (***Figure 7—figure supplement 1B***). Second, these animals had fewer Piwi1⁺ i-cells and numbers of neurons and nematoblasts was lower than normal. RFamide neuron numbers were low or absent altogether (***Figure 7***). Third, these larvae had defects in ciliation (***Figure 7***, ***Figure 7—figure supplement 2***) and, as a result, their movement was restricted (larvae swim by coordinated cilia beat). Finally, the larvae had lost their ability to metamorphose upon CsCl treatment (***Figure 7***), with the latter probably being the result of a loss of neurons expressing GLWamide, the internal regulator of metamorphosis in *Hydractinia* (***Gajewski et al., 1996***; ***Müller and Leitz, 2002***; ***Schmich et al., 1998***). However, the number of S-phase cells was upregulated compared with the control and proliferative cells were also detectable in the epidermis; normally, S-phase cells are restricted to the gastrodermis in larvae (***Figure 7***). The above defects could be explained by *Soxb1* playing a role in maintaining stemness in i-cells, similar to the role of Sox2 in mammalian pluripotency (***Boyer et al., 2005***; ***Takahashi and Yamanaka, 2006***). Loss of *Soxb1* may have resulted i-cells exiting the stem cell program, as observed by Piwi1 staining (***Figure 7***). This would be expected to directly affect all i-cell derivatives that include neurons and nematocytes, as well as other cell types. The increased number of S-phase cells could reflect a compensatory response to loss of i-cells or be a response of i-cells to loss of stemness.

Downregulation of *Soxb2* had no visible effect on i-cells or on GLWamide⁺ neurons (***Figure 7***) but RFamide⁺ neurons were nearly absent. Nematoblast numbers had increased and no significant difference was observed on cycling cells. These phenotypes are markedly distinct from what is seen after downregulation of *Soxb2* in adult neurogenesis, where the numbers of neurons, nematocytes, and cycling cells are noticeably reduced (***Flici et al., 2017***). This could be explained by preferential differentiation of neural progenitors to GLWamide⁺ neurons over RFamide⁺ neurons if progenitors are limiting, given the essential role of the latter in metamorphosis. Alternatively, the SoxB2 protein might be maternally loaded or dispensable for embryonic GLWamide⁺ neurons. The increase in nematoblast numbers following *Soxb2* downregulation could indicate a nematogenesis-inhibiting role for this gene. Surprisingly, downregulation of *Soxb3* had no major effect on the animals, in contrast to the role of this gene in adult neurogenesis (***Flici et al., 2017***). The above data suggest an essential and complex role of *Soxb2* for embryonic neurogenesis, but also highlight major differences between embryonic and adult neurogenesis.

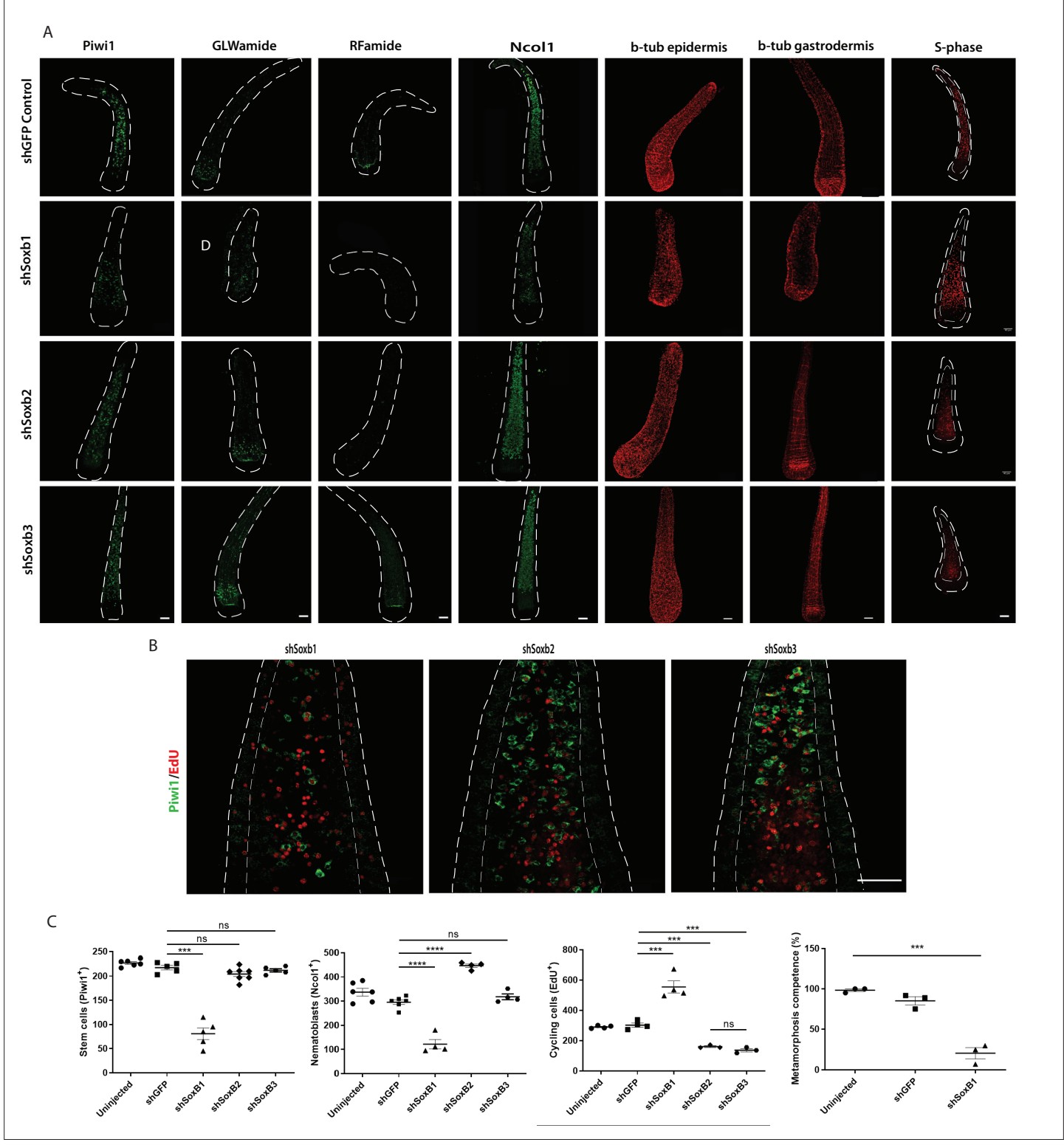

**Figure 7.** Effects of SoxB genes downregulation in embryogenesis. (**A**) Short hairpin RNA (shRNA)-mediated knockdown of *Soxb1*, *Soxb2*, and *Soxb3*. (**B**) Higher magnification of Piwi1⁺ and S-phase cells showing reduced i-cell numbers following *Soxb1* downregulation. Note proliferative cells in epidermis, not seen in untreated and following *Soxb2* or *Soxb3* downregulation. (**C**) Quantification of phenotypes. Scale bars = 40μm.

The online version of this article includes the following figure supplement(s) for figure 7:

**Figure supplement 1.** Validation of shRNA experiments.

*Figure 7 continued on next page*

*Figure 7 continued*

**Figure supplement 2.** Effect of *Soxb1* downregulation on epidermal ciliation in 3-day old larvae using acetylated tubulin antibody staining and in situ hybridization of *Soxb2* in sexual polyps (**A**) shGFP-injected animal.

## Discussion

Multiple SoxB proteins act to generate neurons across metazoans but orthologous relationships between individual SoxB members in different animal clades have been difficult to establish (***Flici et al., 2017***). This is either because multiple duplication events of an ancestral SoxB gene occurred within lineages, resulting in paralogous genes in any one clade, or that sequence drift over evolutionary time has blurred discernable orthologous relationships. However, the requirement for more than one SoxB gene to generate neurons in animals as distantly related as *Hydractinia*, *Drosophila*, and mice suggest that more than one SoxB gene had already been present before the cnidarian-bilaterian split to generate neurons in their last common ancestor; it does not exclude further lineage-specific diversification and gene losses thereafter. Finally, it indicates a highly conserved mechanism of subfunctionization following gene duplication at the base of animals.

*Soxb1* is an i-cell marker and probably fulfills a role in stemness. Its downregulation resulted in loss of i-cells and impacted other aspects of development including the ability to metamorphose, which was expected given that i-cells provide progenitors to multiple lineages. This function of *Soxb1* is similar to the role of vertebrate Sox2. However, vertebrate Sox2 also acts in neural stem cells while in *Hydractinia*, there is no direct evidence so far to support a similar function. Furthermore, a population of long-term, self-renewing neural progenitors has yet to be identified in cnidarians. Our results from *Hydractinia*, along with data obtained from other cnidarians (reviewed by ***Rentzsch et al., 2017***), are also consistent with continuous generation of neurons from multi- or pluripotent cells that also contribute to non-neural lineages, without the existence of a long-term, self-renewing neural stem cell.

*Soxb2* was expressed in a population of cells with uncommitted morphology that differentiated to neural cells, based on GFP retention in neurons and nematocytes in the *Soxb2* reporter animal. Our in vivo analysis of *Soxb1*/*Soxb2* double transgenic reporter animals has shown that *Soxb1*$^+$ cells transform into *Soxb2*$^+$ cells (***Figure 4*** and ***Figure 4—figure supplement 1***). Downregulation of *Soxb2* resulted in neural-specific defects in embryos, consistent with this gene being expressed preferentially in the neural lineage (***Figure 3***). The latter is also supported by the neural transcriptional fingerprint (***Figure 5M***; ***Supplementary file 2***) and single-cell transcriptomic profiles of *Soxb2*$^+$ cells (***Figure 5—figure supplement 3C-E***). Of note, deviation from normal development following *Soxb2* perturbation was not a simple reduction in all neural cells, as seen in adults (***Flici et al., 2017***). Instead, *Soxb2* downregulation caused a loss of RFamide$^+$ neurons but not of GLWamide$^+$ neurons. Because GLWamide$^+$ neurons are essential for metamorphosis of *Hydractinia* larvae (***Plickert et al., 2003***), it is possible that the animals preferentially generate this type of neuron when progenitors are limited. Conversely, nematoblast numbers increased upon *Soxb2* downregulation, suggesting that Soxb2 normally inhibits nematogenesis in embryos, in contrast to its function in adults (***Flici et al., 2017***).

Despite being expressed in embryos and larvae (***Flici et al., 2017***), no visible phenotype was observed following *Soxb3* downregulation, other than the effect of its downregulation in adults (***Flici et al., 2017***). Therefore, *Soxb3* may not play a distinct role in embryonic neurogenesis, or that the effect of its downregulation in embryos is too subtle to be observed by our methods.

i-Cells are concentrated in the lower body column (***Bradshaw et al., 2015***), neural progenitors are scattered in the mid-body column, while most neurons and nematocytes are found in the head. This implies that neural cells differentiate while migrating from the base of the polyp toward the head, reflected by the partly overlapping expression domains of SoxB genes that mark the stages of differentiation. A major question emanating from this work is whether migration is required for

neurogenesis cell autonomously, or whether migrating stem cells become exposed to different micro-environments along the oral-aboral axis that drive their differentiation non-cell autonomously.

The poor conservation of SoxB gene sequences across animals make direct functional comparisons difficult. However, the requirement for multiple SoxB genes and migration seem to be common hallmarks of neurogenesis in all studied animals. Additional common features remain to be discovered. *Hydractinia*'s continuous, predictable, and accessible neurogenesis across all life stages provides an excellent model to address these questions.

# Materials and methods

## Key resources table

| Reagent type (species) or resource | Designation | Source or reference | Identifiers | Additional information |
|---|---|---|---|---|
| Antibody | Anti-Piwi1 (rabbit polyclonal) | https://doi.org/10.1126/science.aay6782 | N/A | IF (1:2000) |
| Antibody | Anti-Piwi2 (Guinea pig polyclonal) | In-house | N/A | IF (1:500) |
| Antibody | Anti-acetylated tubulin (mouse monoclonal) | Sigma-Aldrich | Ca# T7451 | IF (1:1000) |
| Antibody | Anti-RFamide (mouse polyclonal) | https://doi.org/10.1007/s004270050181 | | IF (1:1000) |
| Antibody | Anti-GLWamide (rabbit polyclonal) | https://doi.org/10.1007/s004270050181 | | IF (1:1000) |
| Antibody | Anti-Ncol1 (rabbit polyclonal) | https://doi.org/10.1371/journal.pone.0022725.g001 | | IF (1:500) |
| Antibody | Anti-Ncol3 (guinea pig polyclonal) | https://doi.org/10.1371/journal.pone.0022725.g001 | | IF (1:500) |
| Antibody | Anti-GFP (rabbit polyclonal) | Santa Cruz | Ca# 8334 | IF (1:1000) |
| Antibody | Anti-RFP (rat polyclonal) | Chromotek | Ca# 5F8 | IF (1:1000) |
| Strain, strain background (*Hydractinia symbiolongicarpus*) | 293-10 wild type animals | This paper | N/A | Materials and methods |
| Strain, strain background (*Hydractinia symbiolongicarpus*) | *SoxB1*::tdTomato line | This paper | N/A | Materials and methods |
| Strain, strain background (*Hydractinia symbiolongicarpus*) | *SoxB2*::GFP line | This paper | N/A | Materials and methods |
| Strain, strain background (*Hydractinia symbiolongicarpus*) | *RFamide*::GFP line | This paper | N/A | Materials and methods |
| Strain, strain background (*Hydractinia symbiolongicarpus*) | *RFamide*::*SoxB1*::GFP | This paper | N/A | Materials and methods |

## Experimental model and subject details

### *Hydractinia* culture

Stable clones were grown on glass microscope slides and cultured at 20–22°C in artificial seawater (ASW) under a 14:10 light:dark cycle regime. They were fed four times a week with freshly hatched *Artemia* nauplii and once a week with oysters (pureed when fresh and then stored frozen in aliquots).

Spawning took place approximately one and a half hours after light induction by the release of gametes (sperm and oocytes) in a water column. Once collected, the embryos were stored in 4°C for up to 4 hr to halt their development and provide a wider time frame for injections.

### Stable transgenic reporter animals

Zygotes were injected with plasmid DNA at 3–4 µg/µl concentration. Plasmid was supplemented with 100 mM KCl which helped to reduce mosaic integration. Once the embryo reached the planula larval stage, metamorphosis was induced by a 3–4 hr incubation with 1:5 580 mM CsCl:ASW. Once metamorphosing larvae fully retracted, they were placed on glass slides to settle and form a new colony.

Animals were then bred to produce stable, non-mosaic transgenic offspring. To generate double transgenics, stable second-generation *SoxB1::tdTomato* and *SoxB2::GFP* animals were crossed.

## Method details

### Generation of transgenic animals

Cloning vectors were generated by inserting 5′ upstream regulatory sequence (URS) and 3′ down-stream regulatory sequence of the desired gene (*SoxB1*, *SoxB2,* or *RFamide*) upstream and down-stream of the coding sequence of the fluorescence protein (*GFP* or *tdTomato*), respectively. Cloning and ligation of the fragments was achieved by restriction enzyme-based approach. Chemically competent XL1 Blue *Escherichia coli* bacteria were transfected with the vector and once the plasmid was validated by sequencing, embryos were injected.

For the ectopic expression of SoxB1, the RFamide construct was used and the coding sequence of SoxB1 was inserted between URS and fluorescence protein.

### Cellular staining

Standard immunofluorescence staining was done as previously described in (*Flici et al., 2017*): the tissue was fixed in 4% PFA in PBS for 60 min at room temperature (RT) or overnight (ON) at 4°C followed by three washes of 10 min each with PBS with 0.3% Triton X-100 (PBSTx). For storage, the tissue was dehydrated by incubation in increasing concentrations of ethanol diluted in PBSTx and stored in –20°C (25%, 50%, 75%, and 100%; 5 min each wash). Tissue was then slowly rehydrated by washing in decreasing concentrations of ethanol followed by three washes of 10 min each with PBSTx. The tissue was then blocked for 1 hr in 3% BSA in PBSTx and primary antibodies were added ON at 4°C. The following day, tissue was washed three times 10 min each with PBSTx and blocked again for 15 min with 5% serum in BSA/PBSTx (goat serum unless indicated otherwise). Secondary antibodies were added based on the host of the primary antibodies in 5% serum in BSA/PBSTx for 1 hr in RT and then the tissue was washed three times 10 min each with PBSTx. Nuclear staining was then carried out with Hoechst 33258 (use: 1 in 2000; stock: 20 mg/ml; Sigma-Aldrich; B2883) for 15 min at RT followed by three washes 10 min each with PBSTx. Tissue was then mounted in Fluoroshield (Sigma-Aldrich; F6182) on glass microscopic slides (Fisher Scientific; 11562203).

For the nematocyte capsule staining by FITC-coupled DSA lectin, all the steps described below were performed on the shaker and in RT unless otherwise stated. Fixation was carried out by incubating the tissue in TBS-T for 10–30 min. Following, the tissue was washed three times for 5 min each with TBST-T and blocked for 1 hr with 2% BSA in TBS. Three more washes of 5 min each were carried out with TBS, and then lectin was added in a 15 µg/ml final concentration diluted in TBS-T. Lectin incubation was for 1 hr and samples were protected from light. Three more washes of 5 min each with TBS were carried out, and Hoechst (1:1000) was added for 30 min. Samples were washed again for three times before mounting them in Fluoromount. Mounted samples were left ON at 4°C to cure before imaging. Samples could be imaged only within 2–3 days after curing and could be stored in 4°C – not 20°C.

EdU staining was done as previously described in (*Bradshaw et al., 2015*): all the solutions were prepared according to the manufacturer's instructions. Prior fixation, the polyps were incubated in EdU solution for 30 min or more depending on the experiment (stock concentration 10 mM; 1 µl EdU/1 ml ASW), washed three times in ASW and incubated in $MgCl_2$ before fixation. Polyps are then fixed in 4% PFA in PBS for 1 hr at RT and washed once with 3% BSA in PBSTx for 30 min. Next, two washes with PBSTx were carried out (first wash: 1 hr; second wash: 30 min), followed by two washes with 3% BSA in PBSTx for 5 min each. The tissue was then incubated in Click-iT cocktail for 30 min (~100 µl per 100 polyps; protect from light) followed by three washes of 3% BSA in PBSTx for 20 min each. Following, samples were stained with various antibodies as described above.

### In situ hybridization

#### For conventional FISH

FISH was done as previously described in (*DuBuc et al., 2020*): animals were first fixed for 90 s in ice-cold 0.2% glutaraldehyde (stock: 25%, Sigma-Aldrich; G5882) with 4% PFA (stock: 16%, Alfa Aesar; 43368) in filtered ASW and then in 4% PFA in PBS-0.1% Tween (PTW) for 1 hr at 4°C followed by three

quick washes with PTW. Post-fixation washes were done in increasing concentrations of methanol in PTW and then the samples were either stored for future use or continued with the procedure. Next, the tissue was permeabilized by washes in increasing concentrations of methanol in acetone and rehydrated with washes in decreasing concentrations of methanol in PTW.

The next steps were done with reagents and buffers in RT. Following rehydration of the tissue, three PTW quick washes were done to remove any residual methanol and then the activity of PFA was quenched by two washes 5 min each with glycine (2 mg/ml; Fisher Scientific; BP381-1) in PTW followed by three PTW washes.

The tissue was then pre-hybridized by adding pre-heated hybridization buffer ON, and then hybridized for 2 days in pre-heated hybridization buffer containing the desired probes (1 ng/μl). The hybridization temperature depends on the probe with the range being 55–60°C.

Following, all the post-hybridization washes were done at hybridization temperature. First, the tissue was washed twice with a simpler version of the hybridization buffer for 10 and 40 min. Then, washes for 30 min were carried out in decreasing concentrations of hybridization buffer in 2× SSC followed by three washes of 15 min each in 2× SSC.

At this point, all the washes/incubation steps were moved from hybridization temperature to RT. Quick washes in decreasing concentrations of 0.2× SSC in PTW were done followed by five quick washes in PTW. Once the post-hybridization washes were done, the endogenous peroxidase activity was quenched by incubating the tissue in 3% hydrogen peroxide (Fisher; H/1800/15) in PTW for 1 hr at RT protected from light followed by three washes of PTW for 10 min each. The tissue was then incubated with the first antibody (anti-DIG POD, 1:1000; Roche 11207733910) in 1% blocking buffer (Roche; 11096176001) in maleic acid buffer ON at 4°C followed by three washes of 10 min each with PTW.

The developing process of the first antibody was done preferably with Rhodamine in developing buffer with three incubations of 1 hr each at RT followed by an ON incubation at 4°C.

The next day, at least five PTW washes were carried out, or until the tissue has no residual Rhodamine or Fluorescein, and then the tissue was either prepared for mounting or a second antibody incubation. If the tissue was prepared with a single probe, it was then stained with the nuclear marker Hoechst (1:2000) in PTW for 15 min and washed and mounted in 100 mM Tris-pH 8 in 50% TDE (Sigma; 166782).

In the case of a double FISH, after the ON incubation with the developing buffer, the tissue was washed thoroughly with PTW and then washed once for 10 min with 0.1 M glycine pH2 followed by three quick PTW washes. The tissue was then shortly blocked with 1% blocking buffer in maleic acid buffer and incubated with the second antibody (anti-Fluorescein POD, 1:1000, Roche; 11426346910) in 1% blocking buffer in maleic acid buffer ON at 4°C followed by three washes of 10 min each with PTW.

The developing, post-developing washes, nuclear staining, and mounting were done the same way as described above but instead of Rhodamine, Fluorescein was used. After the post-developing washes but before TDE washes and mounting, the tissue could be quickly washed with increasing concentrations of methanol in PBS and then with decreasing concentrations of the same solutions to remove background staining.

## For SABER FISH

Tissue was fixed and dehydrated the same way like in the conventional FISH as described above. Samples were then incubated in 1% $H_2O_2$ diluted in 100% MeOH (ice cold) for 45 min in RT, followed by two quick washes with ice-cold 100% MeOH. Then, tissues were permeabilized by quick washing in decreasing and then increasing concentrations of methanol diluted in, followed by rehydration with quick washes in decreasing concentrations of methanol in PTW.

The next steps are done with reagents and buffers in RT. Following rehydration of the tissue, three PTW quick washes are done followed by two washes 5 min each with glycine (2 mg/ml; Fisher Scientific; BP381-1) in PTW followed by three PTW washes.

Samples were then washed once for 5 min in triethanolamine (TEA) pH 8, followed by another wash in TEA pH 8 with the addition of 6 μl of acetic anhydrite. Acetic anhydrite would form a drop-like appearance in the well and so the samples were left on the rocker until the drop was dissolved. Another wash with TEA followed but this time 12 μl of acidic anhydrite were added and placed on the rocker until drop was dissolved. A few washes with PTW followed.

Samples were then placed in Whyb buffer (pre-warmed at 43°C) and incubated at 43°C for 10 min. Whyb buffer was then replaced with Hyb1 buffer (pre-warmed at 43°C) and placed in a 43°C incubator ON (pre-hybridization). The next day, the probes in Hybe buffer were added at a concentration of 1 µg/120 µl. Tissue was then incubated with the probes for 2 days at 43°C.

All the following steps were done at Hybe temperature and all the reagents were pre-warmed (43°C). Probes were removed and replaced with Whyb buffer for 10 min. Then, two washes with Whyb buffer for 30 min each were done, followed by a 10 min wash with 50% Whyb buffer in 2× SSCTw. Two more washes with 2× SSCTw were followed for 10 min each.

The steps followed were performed at 37°C. 2× SSCTw was replaced with PTW (two quick washes). Once the samples were warmed up to 37°C, Hyb2/fluor solution was added and samples were incubated for 1 hr at 37°C. Hyb2/fluor solution was also pre-warmed at 37°C before it was added to the samples.

After that, Hyb2/fluor solution was replaced with pre-warmed Whyb2 for a 10 min incubation at 37°C followed by two washes of 5 min each with PTW. Nuclear staining was then performed in RT by diluting Hoechst in PTW (1:2000) and incubating the samples for 45–60 min at RT. Samples were then quickly washed twice with PTW and mounted in 97% TDE. Samples were imaged within 4 days. Three to six animals were imaged per condition.

## shRNA interference

For shRNA interference experiments, primers were designed to clone unique sites of the three SoxB genes. Stock primers (100 mM) containing a T7 site were directly used and incubated at 98°C for 2 min and then left in RT for 10 min to anneal; 2 µl of each primer were used and mixed with nuclease-free water to a final reaction volume of 20 µl. Following, T7 transcription was carried out by using the total reaction volume from the previous step (20 µl) as a DNA template along with 3.5 µl of each NTP (ATP, GTP, UTP, CTP; 100 mM stock concentration), 4 µl of 10× reaction buffer and 3 µl of T7 RNA polymerase mix (HiScribe T7 High Yield RNA Synthesis kit, NEB; E2040). The mixture was then incubated ON at 37°C. The next day, the products were DNase treated for 1 hr at 37°C with 40 µl of DNase solution (5 µl DNaseI in 35 µl RDD buffer; Qiagen RNAse-free DNase set; 79254) and RNA was isolated by using Quick-RNA MiniPrep kit (Zymo Research; R1054) following the manufacturer's instructions. The eluted RNA was quantified using NanoDrop and the quality was checked by running a formaldehyde denature gel. Embryos were injected at a concentration of 250 ng/µl for all three SoxB genes.

## In vivo imaging and analysis

Decapitated polyps from transgenic colonies were transferred to 35 mm imaging dishes with a glass bottom (Ibidi; D 263) and 0.5% low-melt agarose in filtered ASW was placed dropwise on individual polyps in order to secure them as close to the bottom of the dish as possible. Once the polyps had stabilized, low-melt agarose was poured into the dish to cover the whole bottom. On polymerization the dish was transferred to an Andor spinning disc confocal microscope which was used to generate time lapse movies. Images were captured over the duration of the time course as z-stacks at 10 µm apart, using a 10× objective lens. Prior to collecting the Z stack, exposure time and EM gain were set and recorded to the channel of illumination. This was maintained for all experiments. Each specimen was imaged using $\lambda$ 488 nm and $\lambda$ 564 nm lasers. Phase contrast images were also collected as an indication of animal location and viability pre, during, and post imaging. Animals were subjected to a 72 hr imaging with 1 hr imaging intervals. Every hour, a z-stack spanning the whole depth of each animal was obtained in GFP, RFP, and DIC channels. Data were then analysed using ImageJ/Fiji software (ImageJ 1.52i), and two to three focal planes covering the whole cell of interest were selected. In total, 16 animals were imaged.

## Flow cytometry, FACS, and imaging flow cytometry

Tissue dissociation and flow cytometric analysis was done as previously described in *DuBuc et al., 2020* using only feeding polyps. Cell suspensions were labelled with 37.5 µg/ml Hoechst 33342 (Sigma Cat#14533) for 20 min at 18°C. Flow cytometric analysis was performed using a BD FACSCanto II flow cytometer (BD Biosciences, San Jose, CA) which was calibrated according to the manufacturer's recommendations. Data were analysed using Diva v8.0.1 (BD Biosciences) or FlowJo v10 (TreeStar Inc, Olten, Switzerland). Typical gating strategies involved gating on live, nucleated cells (Hoechst

33342 positively stained population) measured using the 405 nm excitation laser line and 450/50 nm photomultiplier tube (PMT) fluorescence detector, followed by doublet exclusion based on forward scatter height versus forward scatter area. For cell cycle analysis an additional doublet exclusion gate was included based on plotting Hoechst 33342 height versus Hoechst area. Expression of the GFP reporter in transgenic animals was measured using the 488 nm excitation laser line in the 530/30 nm PMT fluorescence detector. Cells prepared from wild type animals were used as a gating control and to set the voltages in the cell cycle analysis. An example of a typical gating strategy used in cell cycle analysis as well as representative cell cycle profiles from wild type and transgenic animals' cell cycle profile of distinct cell populations is illustrated in *Figure 5—figure supplements 1 and 2*.

For imaging flow cytometry cell suspensions were prepared, as previously described, from reporter animals: Piwi1::GFP and RFamide::GFP and analysed using INSPIRE software on an Imagestream X Mark II imaging flow cytometer (Luminex Corporation, Northbrook, IL). Analysis was performed using IDEAS software (Luminex Corporation, Northbrook, IL). Samples were gated on focused cells (bright-field gradient root mean squared), cells as measured by brightfied area versus aspect ratio followed by Hoechst 33342 positive population and GFP versus SSC. Hoechst and GFP fluorescence was measured using the 405 nm excitation laser line and 435–505 nm bandpass filter and the 488 nm excitation laser line and 480–560 nm bandpass filter, respectively. Cells isolated from a wild type *Hydractinia* animal were used as a gating control.

For FACS, 100 polyps from each reporter line were dissociated as described above Piwi1::GFP, SoxB2::GFP, and RFamide::GFP (n=3) and re-suspended in 1 ml of filtered ASW containing 0.5% Pronase and 0.1% BSA. FACS was performed on a BD FACSAriaII high-speed cell sorter using a 100 µm nozzle at 20 psi with filtered ASW as sheath. ASW was used as sheath fluid to eliminate osmotic stress to the cells during sorting. GFP-expressing and non-GFP-expressing cell populations were gated on live, nucleated single cells. Cells prepared from matched wild type animals were used as controls. Cell populations were sorted directly into Trizol (Life Technologies, Cat#15596026) for subsequent RNA extraction using the Direct-zol RNA MiniPrep kit. The purity of the sorted populations was checked by sorting directly into ASW followed by flow cytometric analysis on a BD FACSAriaII cell sorter.

## InDrop single-cell transcriptomics

Tissue dissociation was done as described above. Cells were then transferred to 18.5% mannitol in $H_2O$ to replace sea-$H_2O$, which otherwise interferes with critical downstream enzymatic reactions including droplet RT. After filtering of the cell suspension through 100 µm filters, cells were quickly subjected to in-house InDrop microfluidics to generate a single-cell transcriptome dataset as described in *Klein et al., 2015*. Following sequencing of the InDrop library, raw reads were processed using python scripts provided in *Klein et al., 2015*. The resulting genes-versus-cells UMI-filtered count matrix was then processed, analysed, and visualized in R using Seurat 3.0 following standard procedures (*Stuart et al., 2019*). Cluster markers were annotated using BLAST against uniprot KB. Bioinformatics workl-flows and results are provided (*Supplementary file 2*). Raw sequencing reads are available under SRA accession number PRJNA777228.

## Antibody generation

For Piwi2 antibody production, primers containing a 5' HIS-tag were designed to amplify a region of Piwi2. Fragments were cloned and expressed in pET3s expression vector using restriction sites (Nde1 and BamH1 – encoded in primer sequences) and expressed in Rosetta DE3 pLysS bacteria. Protein extract was then analysed and injected into two guinea pigs by Eurogentec. Antisera were analysed by western blots and antibody specificity was validated as shown in *Figure 6—figure supplement 1*.

## Cell type-specific transcriptomes
## Quantification and statistical analysis
### Transcriptome analysis

### Quantitation of shRNAi mutant animals
All cell counting analysis was performed using 3D Object Counter function in ImageJ/Fiji software. Counts were made relative to the total tissue area in the X/Y dimension. A z-stack covering the whole depth of each sample was obtained. Counting was performed on cells positive for the respective

cell type marker based on antibody staining and numbers were compared between shControl and shSoxb1/2/3 animals. Three to seven animals were counted per condition. Uninjected animals and animals treated with shGFP were used as controls. Uninjected animals were only used for quantification purposes. Statistical analysis was performed in GraphPad Prism using unpaired t-test. Differences were considered not significant when p >0.05.

The shRNA knockdown was assessed qualitatively. At least three animals per shRNA treatment were assessed by SABER FISH.

## Software and algorithms

FastQC v0.11.6 (*Babraham Bioinformatics, 2019*)
Trimmomatic v0.36 (*Bolger et al., 2014*) http://www.usadellab.org/cms/?page=trimmomatic
HISAT2 v2.1.0 (*Zhang et al., 2021*) http://daehwankimlab.github.io/hisat2/
HTSeq v0.11.2 (*Anders et al., 2015*) https://htseq.readthedocs.io/en/release_0.11.1/index.html
DESeq2 v1.28.1 (*Love et al., 2014*) https://bioconductor.org/packages/release/bioc/html/DESeq2.html
Seurat 3.1.5 (*Hao et al., 2021*) https://github.com/satijalab/seurat/releases

### Cell type-specific transcriptomes

After RNA isolation, samples were shipped to the NIH Intramural Sequencing Center (NISC) for further processing and sequencing. RNA was amplified with the Ovation RNA-Seq System V2 kit, and sequencing libraries were made with Illumina TruSeq Stranded mRNA Library Prep Kit. Libraries were sequenced on two lanes of Illumina NovaSeq 6000 (2 × 151 bp), generating between 27 and 69 million reads per sample (average 44 million reads).

### Transcriptome analysis

Read quality control was performed using FastQC v0.11.6. Overrepresented sequences and low-quality bases were trimmed using Trimmomatic v0.36 (*Bolger et al., 2014*). After trimming, unpaired reads and reads shorter than 36 bp were discarded.

Reads were aligned to the reference transcriptome described in *DuBuc et al., 2020* using HISAT2 (*Kim et al., 2019*), and gene-level read counts were generated with HTSeq-count v0.11.2 (*Anders et al., 2015*). Cell type-specific transcripts were identified by conducting a differential expression analysis between reporter-positive and reporter-negative cells using the R package DESeq2 (*Love et al., 2014*).

## Acknowledgements

We thank Áine Varley for animal care and the NIH Intramural Sequencing Center (NISC) for generating the sequence data. Anti-minicollagen antibodies were a kind gift from Suat Özbek (University of Heidelberg). Confocal images were taken at the Centre for Microscopy and Imaging Core Facility at NUI Galway. All flow cytometry and imaging cytometry analyses were performed in the Flow Cytometry Core Facility at NUI Galway. This work utilized the Biowulf high-performance supercomputing resource of the Center for Information Technology at the National Institutes of Health (https://hpc.nih.gov).

## Additional information

### Funding

| Funder | Grant reference number | Author |
| --- | --- | --- |
| Wellcome Trust | 210722/Z/18/Z | Uri Frank |
| Science Foundation Ireland | 11/PI/1020 | Uri Frank |

| Funder | Grant reference number | Author |
|---|---|---|
| National Science Foundation | 1923259 | Uri Frank<br>Christine E Schnitzler |
| National Human Genome Research Institute | ZIA HG000140 | Andreas D Baxevanis |
| Science Foundation Ireland | 13/SIRG/2125 | Sebastian G Gornik |
| Human Frontiers Science Program | LT000756/2020-L | Miguel Salinas-Saavedra |

The funders had no role in study design, data collection and interpretation, or the decision to submit the work for publication. For the purpose of Open Access, the authors have applied a CC BY public copyright license to any Author Accepted Manuscript version arising from this submission.

## Author contributions

Eleni Chrysostomou, Conceptualization, Formal analysis, Investigation, Methodology, Writing – original draft, Writing – review and editing; Hakima Flici, Sebastian G Gornik, Miguel Salinas-Saavedra, James M Gahan, Emma T McMahon, Shirley Hanley, Investigation; Kerry Thompson, Michelle Kilcoyne, Methodology; Christine E Schnitzler, Andreas D Baxevanis, Data curation, Formal analysis; Paul Gonzalez, Formal analysis; Uri Frank, Conceptualization, Formal analysis, Funding acquisition, Investigation, Project administration, Supervision, Writing – original draft, Writing – review and editing

## Author ORCIDs

Eleni Chrysostomou http://orcid.org/0000-0003-2455-6321
Sebastian G Gornik http://orcid.org/0000-0002-8026-1336
Miguel Salinas-Saavedra http://orcid.org/0000-0002-1598-9881
Emma T McMahon http://orcid.org/0000-0003-3933-8853
Kerry Thompson http://orcid.org/0000-0003-2721-8977
Christine E Schnitzler http://orcid.org/0000-0002-5001-6524
Andreas D Baxevanis http://orcid.org/0000-0002-5370-0014
Uri Frank http://orcid.org/0000-0003-2094-6381

## Decision letter and Author response

Decision letter https://doi.org/10.7554/eLife.78793.sa1
Author response https://doi.org/10.7554/eLife.78793.sa2

---

# Additional files

## Supplementary files

- Supplementary file 1. Combination of cell fractions used for differential gene expression analysis.
- Supplementary file 2. Differentially expressed gene lists.
- Supplementary file 3. Files pertaining to single-cell transcriptomic analysis.
- Supplementary file 4. List of SRA accession numbers.
- MDAR checklist

## Data availability

The accession number for the RNA-seq datasets generated in this study is Sequence Read Archive (SRA): BioProjects PRJNA549873 (bulk RNA-seq) and PRJNA777228 (single cell RNA-seq). Accession numbers for each sample are listed in Table S2. The Hydractinia symbiolongicarpus genome is available through the National Human Genome Research Institute of the US National Institutes of Health (https://research.nhgri.nih.gov/hydractinia/). Corresponding data is archived in the NCBI Sequence Read Archive (SRA) under BioProject PRJNA807936.

The following datasets were generated:

| Author(s) | Year | Dataset title | Dataset URL | Database and Identifier |
|---|---|---|---|---|
| Chrysostomou | 2022 | Bulk RNA-seq | https://www.ncbi.nlm.nih.gov/bioproject/PRJNA549873/ | NCBI BioProject, PRJNA549873 |
| Frank U | 2022 | Single-cell RNA-seq | https://www.ncbi.nlm.nih.gov/bioproject/PRJNA777228 | NCBI BioProject, PRJNA777228 |

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
