## [Editor Report]

This paper shows that SoxB genes act sequentially in neural stem cells before differentiating into mature neural cells and that loss of SoxB genes causes defects in embryonic neurogenesis. The manuscript provides molecular insights into neurogenesis during cnidarian embryogenesis, homeostasis, and regeneration.

---

## [Decision Letter]

[Editors' note: this paper was reviewed by Review Commons.]

---

## [Author Response]

Reviewer #1 (Evidence, reproducibility and clarity (Required)):The authors present further investigation of the Sox transcription factors in the model Cnidarian Hydractinia. They showcase the Hydractinia as now a relatively technically advanced model system to study animal stem cells, regeneration and the control of differentiation in animal cells. In this study they characterise the neural cells in hydractinia using FACS and sing cell transcriptome sequencing, investigate the sequential expression of SoxB genes in the i-cells and presumptive lineage giving rise to i-cells and investigate the neuronal regeneration making good use of transgenic rules. Finally, they investigate the role of SoxB genes in embryonic neurogenesis.There are no major or minor issues effecting the conclusionsReviewer #1 (Significance (Required)):This study helps to confirm the role of an important group of transcription factors is conserved across the metazoan as well as showcasing an exciting model organism for regeneration and stem cell biology. This will of interest to a broad audience of developmental and biologists.My own research is in the same field, using a different model systemReferees cross-commentingI agree with the comments from the other reviewers, and am sure the authors can address these adequately with further explanation.Reviewer #2 (Evidence, reproducibility and clarity (Required)):SummaryChrysostomou et al. investigate the role of three putative SoxB genes in embryonic neurogenesis in the colonial hydrozoan Hydractinia. They show that SoxB1 is co-expressed with Piwi in the multipotent i-cells and, using transgenics, they show that these Piwi/SoxB1 cells become neurons and gametes, consistent with the cell types that differentiate from i-cells. They further suggest that SoxB2 and SoxB3 are expressed downstream of SoxB1 in the progeny of the i-cells and, using shRNAs, investigate the role of SoxB genes on embryonic neurogenesis. The primary conclusions center on the similarity between neural differentiation in humans and Hydractinia as both systems pattern neurons using sequential expression of SoxB genes during the differentiation of neurons. The manuscript presents a large and diverse set of data derived from analysis of transgenic animals, single-cell sequencing, and investigation of gene function; despite this, the conclusions are either not particularly novel or not well-supported. The co-expression of SoxB1 in Piwi-expressing i-cells appears to be both novel and significant but the implications are not clearly indicated. Additional specific concerns are detailed below.Major comments1. SoxB genes act sequentiallyKnockdown of SoxB2 has already been shown to result in the loss of SoxB3, so the sequential action of SoxB genes in this animal does not seem to be a terribly novel conclusion.

Sequential expression of Soxb1-Soxb2 has not been demonstrated previously. Flici et al. did show some data on Soxb1 expression but these were not detailed. Furthermore, they have not shown in vivo transition to Soxb2. Our new single-molecule fluorescence in situ hybridization, and the transgenic reporter animals have been developed to address these issues.

While this manuscript does appear to report the most comprehensive analysis of SoxB1 expression, theevidence for sequential activation of SoxB1 and then SoxB2 in the same lineage (Figure 4) is a bit troubling. Panel A of this figure appears to show complete overlap between SoxB1 and SoxB2, suggesting all the cells in this field are synchronously passing through the transition point from SoxB1 to SoxB2 expression. While this may reflect reality, it would be more convincing to see adjacent cells expressing SoxB1 only or SoxB2 only, reflecting the dynamic progression of cell type specification along the main body axis.

As shown in Figures 1, Soxb1 is expressed by i-cells (together with Piwi1) in the lower body column of feeding polyps and in germ cells in sexual polyps. These cells do not express Soxb2. Figure 2 shows that Soxb2 is expressed more orally in a population of putative i-cell progeny as they migrate towards the head. These cells still express Soxb1. In the upper part of the body column, just under the tentacle line, there are Soxb2+ cells that do not express Soxb1. Therefore, cells expressing Soxb1 but not Soxb2 are present in the basal part of the polyp, Soxb1+/Soxb2+ double positive cells in the mid body region (i.e., the interface between the two domains where Soxb1+ cells start to express Soxb2 and downregulate Soxb1.), and cells expressing Soxb2 but not Soxb1 in the upper part of the polyp, just under the tentacle line.

In Figure 4, we show the interface between these two domains using in vivo imaging of double transgenic reporter animals to visualize the Soxb1 to Soxb2 transition. Indeed, in the mid body area, most Soxb1+ cells also express Soxb2 (Figure 2). Hence, Figure 4 should be seen keeping Figure 2’s data in mind. At the mRNA level, the overlap between the Soxb1 and Soxb2 domains is smaller (Figure 2) than the one shown in Figure 4 because the latter constitutes a lineage tracing, showing fluorescent proteins with a long half-life. Therefore, when i-cells downregulate Soxb1 while starting to express Soxb2, the long half-life of tdTomato results in red fluorescence persisting longer than the mRNA encoding it. We have added cartoons to Figure 4 to indicate the position along the main body axis that are depicted.

Panel B is more concerning; while the authors have highlighted a cell that does appear to transition from SoxB1+ to SoxB1+/SoxB2+, there are several cells in the background that appear to gain SoxB2 expression without first expressing SoxB1. Do these cells constitute a fundamentally different, SoxB1-indpenendent, lineage of SoxB2+ cells? This would be noteworthy but is not mentioned or characterized.

The panels included in Figure 4 constitute selected confocal slices of stacks acquired in vivo. During imaging, cells move in three dimensions, making them appear and disappear in given optical planes over time. In other words, the individual time frames shown (T0-T5) were not always found in the same plane due to cell migration in the Z dimension. The cells that appear to gain Soxb2+ w/o having expressed Soxb1 first are an example of such cells. They are probably Soxb2+ cells that had already downregulated Soxb1 and migrated into the respective plane of image. We have added the explanation to Figure 4's legend.

Figure 7 shows the effect of SoxB1 knockdown (by shRNA) on the number of Piwi-expressing cells, nematocytes, etc but why not show that SoxB2 and SoxB3 are also knocked down in these experiments? Figure S11 shows no effect of SoxB2 and SoxB3 knockdown on SoxB1 expression but why wasn't the reciprocal experiment performed? If SoxB2 and SoxB3 are really downstream of SoxB1, the authors should demonstrate that with the shRNA experiments.

Our data show that Soxb1 is expressed in i-cells and its KD reduces the number of these stem cells (assessed by expression of Piwi1, an i-cell marker). Because i-cells give rise to all Hydractinia somatic lineages (and to germ cells), focusing specifically on Soxb2+ cells would provide no further insight because all cell types are expected to be affected. Indeed, injection of shRNA targeting Soxb1 resulted in smaller animals with multiple defects, including but not limited to the neural lineage.

2. Knockdown of SoxB genes resulted in complex defects in embryonic neurogenesisThe manuscript aims to detail the roles of SoxB1, SoxB2, and SoxB3 in embryogenesis but only one of the main figures even shows pre-polyp life stages (Figure 7) and the results presented in in this figure are confusing. The authors should provide additional support to show that this reagent is working as expected.

This information is included in Figure S11. Using mRNA in situ hybridization, we show that injection of shRNA targeting Soxb3 causes transcriptional downregulation of Soxb3 but not of Soxb2. The figure also shows the specificities of the shRNAs targeting Soxb1 and Soxb2.

Further, the results for SoxB1 and SoxB2 knockdown do not support the previous investigation of the role of SoxB2 in neurogenesis (Flici et al. 2017). If SoxB1 is upstream of SoxB2, how does knockdown of SoxB1 have such a dramatic effect on RFamide neurons and nematocytes but knockdown of SoxB2 has an effect only on RFamide neurons? Is it possible the SoxB2 shRNA also wasn't working as expected? Can the results of the Flici et al. 2017 paper showing SoxB2 knockdown in polyps be recapitulated using these shRNAs? If the point is to argue that embryos and adults (polyps) use fundamentally different mechanisms to drive neurogenesis, then the results presented in Figures 1-6 (which investigate SoxB genes in polyps) can't really be used to make inferences about embryonic neurogenesis. I think the authors have more work to do to demonstrate that embryonic and adult neurogenesis fundamentally differ.

The Soxb2 shRNA specificity is shown in Figure S11 (i.e., it KD Soxb2 but not Soxb1). We were equally surprised to discover that Soxb2 KD resulted in somewhat different phenotypes than the ones obtained by Flici et al. (2017) in polyps. At this stage, we cannot explain the difference. However, one could speculate that it resulted from slightly different regulation logic between embryonic and adult neurogenesis. More specifically, we propose different priorities for generating neural subtypes as explanation.

Unfortunately, shRNAs work only with embryos, and long dsRNA mediated KD works only with polyps. CRISPR/Cas9-mediated KO is feasible in Hydractinia, but knocking out developmental genes, such as these Sox genes, would likely cause embryonic lethality. Other conditional KO/KD approaches are not available for Hydractinia.

We believe we have made all possible efforts to clarify the roles of these genes using currently available techniques. Neurogenesis is a complex process that is only partially conserved among different animals and poorly studied in non-bilaterians. Furthermore, it is not possible to answer all questions in one study. As many studies before, our work contributes to the understanding of neurogenesis but also raises new questions. Addressing them is matter for future research. We have toned down the statement in the last sentence of the results and in the discussion and do not claim that embryonic and adult neurogenesis are fundamentally different.

Minor commentsMethods: A large bit of data from this manuscript relies on quantitative analysis of cell number but there's not enough information in the methods to understand how quantification was performed. How many slices from the z-stack were analyzed? Were counts made relative to the total tissue area in the X/Y dimension or relative to the number of total nuclei in the same section? How many individuals were examined for each analysis?

All cell counting analysis was performed using ImageJ/Fiji software. Counts were made relative to the total tissue area in the X/Y dimension (for the shRNA experiments). A Z-stack covering the whole depth of each larva was obtained. Counting was performed on cells positive for the respective cell type marker based on antibody staining and numbers were compared between shControl and shSoxb1/2/3 animals. At least 4 animals were counted per condition.

Page 11 – "Piwi2low cells, which are presumably i-cell progeny" – how were "high" and "low quantified?

“High” and “low” were not quantified. This is because i-cells progressively downregulate Piwi genes (i.e., Piwi1 and Piwi2) as they differentiate but this is a continuous process. Hence, it is difficult to put a threshold of Piwi1/Piwi2 protein level below which a cell ceases to be an i-cell while becoming a committed progeny. This is a similar process that is well documented in other animals where stemness markers are gradually downregulated during differentiation.

Page 13 – "a role in maintaining stemness" – this comment is not totally clear to me. Why would the number of EdU+ cells increase if the role of SoxB1 is to maintain stemness? Wouldn't SoxB1 knockdown then force stem cells to exit their program, resulting in early differentiation of i-cell progeny? This should be clarified.

KD of Soxb1 resulted in a decrease in the number of i-cells (i.e., Piwi1+ ones), suggesting that the gene is required for stemness maintenance. The increase in the numbers of cells in S-phase in this context was not related to i-cells because most of them were Piwi1-negative (Figure 7B). The identity of the cells in S-phase remains unknown, but a plausible explanation is that i-cell progeny (e.g., nematoblasts; see also next comment) increase their proliferative activity when i-cells numbers are low as a compensatory mechanism. This is merely a speculation. We have rephrased the paragraph to increase clarity.

Page 13 – "if progenitors are limiting" – if progenitors are limited why would there be an increase in nematocytes?

We do not have a definitive answer to this question but speculate that nematoblasts (i.e., stinging cell progenitors) account, at least in part, for the excessive proliferation seen under Soxb1 KD. This may constitute a mechanism allowing a depleted i-cell population to recover by self-renewal (instead of differentiation), moving temporarily the proliferation task to committed progeny (e.g., nematoblasts) until i-cell numbers return to normal. However, in the absence of evidence we refrain from expanding on this in the text.

Figures 1 and 2 claim to show "partial overlap" but they look perfectly overlapping to me. This makes the situation in Figure 4B difficult to interpret.

Figure 1 shows full overlap between Piwi1 and Sox1 expression and this is reflected in the text. Figure 2 shows no overlap between Soxb1 and Soxb2 in the lower body column (where only Soxb1 is expressed), overlap in the mid body region, and Soxb2 only expressing cells in the upper part of the body, just under the tentacle line. Similarly, the figure shows overlap between Soxb2/Soxb3 under the tentacle line, and predominantly Soxb3 above it in the head region. The small cartoons at the left side of each panel indicate its position along the oralaboral axis. See also our reply to the second part of comment #1.

Figure 4 – No indication of which part of the animal or which stage is shown in these images.

We have added cartoons to indicate the area in the polyp from where the images were taken.

Figure 5 – No indication of where these dissociated cells came from – polyps? Larvae?

All tissue samples were taken from feeding polyps; this is now mentioned in the Materials and methods section.

Panel D is a bit perplexing – what are the "progeny" of Piwi+ cells if not SoxB2+ cells and their derivatives?

In Panel D, we show three cell fractions. One constitutes i-cells, based on high Piwi1 expression (green fluorescence of the Piwi1::GFP reporter transgene) and morphology; one fraction includes nematocytes, based on the characteristic nematocyst capsule, and one constitutes a mixture of other i-cell progeny. The latter includes different cell types, given that i-cells are thought to contribute to all lineages. They have only dim GFP fluorescence because the Piwi1 promoter-driven GFP shuts down upon i-cell differentiation. Soxb2+ cells are also among them but are not the only i-cell progeny.

Why are nematocytes but not neurons indicated?

Neurons are shown on Panels E and F. See also next comment.

Piwi seems to be maintained in Ncol-expressing cells but not in SoxB2- or RFamide-expressing cells? Does this suggest that Piwi is turned on in i-cells, off in SoxB2-expressing cells, and on again in terminally differentiating nematocytes? This would be quite surprising and should be verified with antibody labeling/imaging in Piwi transgenics to confirm the result. The resolution for Panel M is too low to evaluate this part of the figure.

The Piwi1i gene is downregulated upon i-cell differentiation. In the Piwi1:GFP reporter animal, residual GFP fluorescence persists post differentiation due to GFP's long half-life. The brightness of which depends on the time elapsed since differentiation. Because nematocytes are short living cells with high turnover, most nematocytes have recently differentiated and are therefore relatively bright green in the Piwi1::GFP animal. Neuron turnover is lower, making most neurons in the same transgenic animal appear dim. The resolution of the imaging flow cytometer is limited because the machine images 1000s of cells per second through all optical channels. However, it is high enough to allow the identification of features such as cell shape, some organelles (e.g., nematocytes), nuclear size and shape, and fluorescence intensity.

Figure 7 – the low magnification images provide nice overall context but the authors should also provide high magnification panels for the same images. Without them it is not possible to assess "defects in ciliation" or to determine if there are defects in GLWamide neurons from these knockdowns (e.g., neurite vs cell body defects). There's no mention of the fact that SoxB1 knockdown resulted in complete loss of RFamide cells, which is strange. Are there SoxB2independent populations of RFamide? Panel B could be interpreted multiple ways – downregulation of Piwi in SoxB1 shRNA or upregulation in SoxB2/B3. The authors should provide an image of control shRNA-injected larvae with the same co-labeling of Piwi/EdU for context. From the images, it's not clear that there were differential effects of SoxB2 and SoxB3 on nematocytes.

The resolution of the images is, in fact, high, allowing it to be blown up on the screen. Even higher magnification of ciliation can be seen in Figure S12. KD of Soxb1 resulted in complete or nearly complete loss of Rfamide+ neurons. We have added this statement to the text as requested. Panel B shows the relative difference in Piwi1+ and S-phase cells between shSoxb1, shSoxb2, and shSoxb3-treated animals. The quantification relative to the control is presented in Figure 7C.

Figures 6 and S9 – why piwi2 and not piwi1?

In Figure 6, we co-stained the regenerates with two antibodies: one was a rabbit anti-GFP (to visualize the RFamide+ neurons), and the other was a guinea pig anti-Piwi2 (to visualize icells). The anti-Piwi1 antibody that was used in other images to visualize i-cells was raised in rabbit and could not be used in conjunction with the anti-GFP one.

Figure S1 – Kayal et al. 2018 is the most recent phylogeny of cnidarians and should probably be cited in place of Zapata throughout the manuscript. Independent of this, the polytomy in Figure S1 panel A is not supported by either Zapata or Kayal and should be fixed.

We have cited Kayal et al. 2018 and revised the tree in Figure S1 as pointed.

Figure S3 – is this mRNA? Protein? Panels E-G are too small to interpret. Please provide stage/time for cartoons in panel H.

As per the legend, Panels A, B, D, E, F refer to protein; C is lectin staining (DSA), and G is EdU. The resolution of Panels E-G is actually high, allowing blowing up of the images on the screen to view the details. The stages of the cartoon in Panel H are now provided in the figure legend.

Figure S11 – please provide images of whole larvae as shown for Piwi knockdown in Figure S9 and some additional support (e.g., qPCR) to demonstrate the shRNAs are actually working.

Figure S9 represents immunostaining using the anti-Piwi1 antibody. In Figure S11, we show the specificity of the shRNA treatments; we used highly sensitive single-molecule mRNA in situ hybridization. Whole animal imaging is not informative due to the punctuated nature of the single-molecule staining.

Figure S12 – it's not clear what ciliary "defects" are being shown.

In the control, cilia are uniformly distributed along the oral-aboral axis whereas in the shSoxb1-injected animals, the pattern is patchy. Additionally, shSoxb1-injected larvae could not swim (planulae swim by coordinated cilia beat).

Reviewer #2 (Significance (Required)):Generally, the results are either equivocal or the conclusions are not well supported by the results (as detailed above). The significance of this work to vertebrate neurobiology is somewhat weak. (Especially considering the orthology of these genes to bilaterian SoxB genes is not well supported.) Why not compare these results to other cnidarians – the expression patterns of SoxB1 and SoxB2 in corals and sea anemones seem to differ quite a lot (Shinzato et al. 2008; Magie et al. 2005), suggesting these genes are almost certainly not behaving in the same way across cnidarians. This is exciting! What's happening in Hydra? Seems like it should be possible to mine the single-cell data set from Siebert et al. to test these hypothesized relationships between the Sox genes in another hydrozoan which constantly makes new neurons.

We have modified the concluding section in the discussion, in line with this comment. See also comment to Reviewer #3.

Reviewer #3 (Evidence, reproducibility and clarity (Required)):This paper characterizes the role of Soxb genes in neurogenesis in Hydractinia. The authors use cutting edge approaches including FISH, transgenics, image flow cytometry, FACS and shRNA knock downs to characterize SoxB in Hydractinia. The images are beautiful, the data is sound and the interpretation of the data is appropriate.I have only minor suggested listed by section below:Abstract– The abstract and introduction should make clear that this is a colonial animal and the cell migration occurs from the aboral to the oral end of the polyp (not the animal, as there are many oral ends). This is relevant to the interpretation of the data as the polyps do not act in isolation as they interconnected and may communicate via the stolonal network that connects the polyps in the colony.

We have added a section to the Introduction to address the reviewer's comment. The Abstract, however, is too short to include this explanation.

– The human disease justification is a relatively weak one and does not need to be included. Using Hydractinia to understand the role of SoxB in the evolution of neurogenesis in animals is enough justification for the study.

We have adopted the reviewer's comment and modified the statement in the discussion (see also comment to Reviewer #2).

Introduction– Instead of Sox phylogenies (the term phylogeny is more appropriate for species trees), consider substituting, for Sox gene trees. And instead of "phylogenetic relation" use the term "orthology"

This has been done.

– The number of times the sentences that have the sentiment "….remain unknown." "….little is known.." "…unclear…" , "….difficult to establish…." etc. is distracting and detracts from what IS known about these genes. It is not necessary to continually justify the study throughout the introduction. Instead a clearer description of the background and setting up the question/hypothesis of SoxB paralog subfuctionalization in space and time – would be more informative to the reader.

We have reduced the number of occasions as recommended.

– The authors state that there are three SoxB genes in the Hydractinia genome? What genome? For several years there has been multiple papers published by subsets of these authors have used unpublished genome data, but the complete genome has yet to be released to the public. This is especially egregious because they cite their NSF funded EDGE proposal to CEF and UF which is supposed to develop tools to the community, and yet the community at large doesn't have access to the genome. If these data came from the genome, then the genome should be released. If these data came from a previously published transcriptome as in the previous SoxB paper then this should be stated explicitly.

The Hydractinia genome assembly, annotation, RNA-seq data, and genome browser are now available in the Hydractinia genome project portal at the National Human Genome Research Institute (NIH) website https://research.nhgri.nih.gov/hydractinia/. The raw data have been deposited in the NCBI Sequence Read Archive (SRA) under BioProject PRJNA807936. This information has been added to the 'Resource availability' section.

Results– I assume there was no expression of Soxb2 and Soxb3 in the reproductive polyps? This should be stated explicitly.

Soxb2 expression in sexual polyps was consistent with the nervous system and with maternal deposition in oocytes. It was not detected in male germ cells. We have added a new in situ hybridization image of Soxb2 to Figure 12.

– The word "progeny" is used throughout to describe terminally differentiated cells. However, progeny implies offspring, but these are actually later stages of differentiation of the in a cell's ontogeny, thus the term should be changed to "differentiated cells"

We used "progeny" to indicate that the corresponding cells derived from a specific progenitor cell type. We did try replacing it with "differentiated cells" but this completely changes the meaning of the sentence: first, it does not include the cell of origin info and second, not all progeny are already fully differentiated.

– Typo on page 11 "This predictable generation of many new neurons provides an opportunity to study neurogenesis in [a ]regeneration." – Remove the "a"

Corrected.

– While the regeneration study is interesting, there is nothing revealed about the role of Soxb and there is not a lot of new information revealed about regenerations. Authors should better justify this section or consider omitting.

These sections demonstrate de novo neurogenesis in head regeneration. This was not known in this animal before.

Discussion– The authors assume that in the transgenic lineage, the fluorescent marker in differentiated cells is due to retention of fluorescence, but it is unclear if they can rule out that Soxb2 is still being expressed in those cells" Please clarify.

We conclude this by comparing the mRNA expression (Figures 1 and 2) with the fluorescent proteins (Figure 3).

– How did the authors determine that the shSoxb3 knockdown worked? Please discuss relevant controls and validation (either in discussion or methods). This is particularly important given that it didn't have an apparent phenotypic effect.

The efficacy of all shRNAs determined by in situ hybridization, showing that each shRNA downregulates its own target mRNA but not the others (Figure S11).

– Again, the connection to human health is a bit of a stretch. Instead, what is most interesting is the similarity of Soxb paralogs acting sequentially as has been found in vertebrates. This suggests a highly conserved mechanism of subfunctionization following gene duplication at the base of animals.

We agree. This is now also better highlighted in the discussion.

Figures– Its very hard to distinguish the overall abundance of Soxb2 and Soxb3 expression along the polyp body axis from the panels figure 2. A lower magnification or larger area in each region would be helpful.

In Figure 2, we performed single-molecule in situ hybridization. While highly sensitive, this method generates spotty images because they highlight single molecules and are not coupled to an enzymatic reaction as in other methods. They mostly looks poor when showing low magnification images. Because a previous study (Flici et al. 2017) has already shown the general expression pattern, we aimed at providing the details of the transition.

– Figure 4 – either the figure is upside down or the text is upside down. It is also difficult to see the double staining (if any).

The figure is oriented to position the oral end up. The resolution of the panels is high, enabling blowing-up on the screen. The quality of in vivo time lapse images cannot match that of fixed and antibody stained ones, or of single in vivo images. This is because the animals are imaged for many hours during which they tend to bleach.

*–* Figure 5M is difficult to read due to the small print. Consider enlarging and moving it to Supplementary Material.

The size of the text is small but the resolution is very high, enabling blowing up the image on the screen. We thought that the information was important enough to be presented in the main text and given that most readers would use the electronic version we preferred this option on another supplemental figure on top of the 12 we already have.

Reviewer #3 (Significance (Required)):This is an interesting and important study because although it is well known that SoxB genes function in neurogenesis in animals, it is unclear how and if subfunctionalization occurs outside of vertebrates. Hydractinia is an excellent model to study SoxB genes because of its colonial organization and continuous development of nerve cells throughout the life of the animal. In addition, it is part of the early diverging cnidarian lineage and thus can provide insight into the relative conservation of SoxB genes across animals.